# Shedding light on blue-green photosynthesis: A wavelength-dependent mathematical model of photosynthesis in *Synechocystis* sp. PCC 6803

Tobias Pfennig [1,2]*, Elena Kullmann[1], Tomáš Zavřel [3], Andreas Nakielski [1,4], Oliver Ebenhöh [2,5], Jan Červený[3], Gábor Bernát [6], Anna Barbara Matuszyńska [1,2]*

1 Computational Life Science, Department of Biology, RWTH Aachen University, Aachen, Germany,
2 Cluster of Excellence on Plant Sciences, Heinrich Heine University Düsseldorf, Düsseldorf, Germany,
3 Department of Adaptive Biotechnologies, Global Change Research Institute, Czech Academy of Sciences, Brno, Czechia, 4 Institute for Synthetic Microbiology, Heinrich Heine University Düsseldorf, Düsseldorf, Germany, 5 Institute of Theoretical and Quantitative Biology, Heinrich Heine University Düsseldorf, Düsseldorf, Germany, 6 Aquatic Botany and Microbial Ecology Research Group, HUN-REN Balaton Limnological Research Institute, Tihany, Hungary

* tobias.pfennig@rwth-aachen.de (TP); anna.matuszynska@cpbl.rwth-aachen.de (AM)

**Data Availability Statement:** All materials are available at https://github.com/Computational-

## Abstract

Cyanobacteria hold great potential to revolutionize conventional industries and farming practices with their light-driven chemical production. To fully exploit their photosynthetic capacity and enhance product yield, it is crucial to investigate their intricate interplay with the environment including the light intensity and spectrum. Mathematical models provide valuable insights for optimizing strategies in this pursuit. In this study, we present an ordinary differential equation-based model for the cyanobacterium *Synechocystis* sp. PCC 6803 to assess its performance under various light sources, including monochromatic light. Our model can reproduce a variety of physiologically measured quantities, e.g. experimentally reported partitioning of electrons through four main pathways, $O_2$ evolution, and the rate of carbon fixation for ambient and saturated $CO_2$. By capturing the interactions between different components of a photosynthetic system, our model helps in understanding the underlying mechanisms driving system behavior. Our model qualitatively reproduces fluorescence emitted under various light regimes, replicating Pulse-amplitude modulation (PAM) fluorometry experiments with saturating pulses. Using our model, we test four hypothesized mechanisms of cyanobacterial state transitions for ensemble of parameter sets and found no physiological benefit of a model assuming phycobilisome detachment. Moreover, we evaluate metabolic control for biotechnological production under diverse light colors and irradiances. We suggest gene targets for overexpression under different illuminations to increase the yield. By offering a comprehensive computational model of cyanobacterial photosynthesis, our work enhances the basic understanding of light-dependent cyanobacterial behavior and sets the first wavelength-dependent framework to systematically test their producing capacity for biocatalysis.

Biology-Aachen/synechocystis-photosynthesis-2024.

**Funding:** This work was funded by the Deutsche Forschungsgemeinschaft (https://www.dfg.de/) under Germany´s Excellence Strategy – EXC-2048/1 – project ID 390686111 (TP, OE, ABM); Deutsche Forschungsgemeinschaft Research Grant - project ID 420069095 (EK, AN, ABM); Deutsche Forschungsgemeinschaft FOR 5573—GoPMF - project number 507704013 (ABM); Ministry of Education, Youth and Sports of CR (grant number LUAUS24131) within the CzeCOS program (https://www.msmt.cz/) (grant number LM2018123), under the OP RDE (https://commission.europa.eu/) (grant number CZ.02.1.01/0.0/0.0/16 026/0008413 'Strategic Partnership for Environmental Technologies and Energy Production') (TZ, JC); as well as by the National Research, Development and Innovation Office of Hungary, NKFIH (https://nkfih.gov.hu/) (awards K 140351 and RRF-2.3.1-21-2022-00014) (GB). The funders had no role in study design, data collection and analysis, decision to publish, or preparation of the manuscript.

**Competing interests:** The authors have declared that no competing interests exist.

## Author summary

In this study we developed a computer program that imitates how cyanobacteria perform photosynthesis when exposed to different light intensity and color. This program is based on a mathematical equations and developed based on well-understood principles from physics, chemistry, and physiology. Mathematical models, in general, provide valuable insight on the interaction of the system components and allow researchers to study complex systems that are difficult to observe or manipulate in the real world. We simulate how energy captured through photosynthesis changes under different lights. We also hypothesize how the production capacity is changed when cells are exposed only to a monochromatic light. By understanding how cyanobacteria react to different lights, we can design better experiments to use them for the production of various products.

## Introduction

Cyanobacteria are responsible for a quarter of global carbon fixation [1]. They are, in fact, the originators of oxygenic photosynthesis, later transferring this capability to other organisms via endosymbiosis [2]. Despite their relative simplicity in cellular structure, the cyanobacterial photosynthetic machinery is a highly sophisticated system that shows significant differences from their plastidic relatives [3]. Recently, they have emerged as a powerful resource for research and biotechnology due to their unique combination of beneficial traits and photosynthetic capabilities [4]. In the quest for environmentally friendly alternatives to fossil fuels and sugar-based production, cyanobacteria stand out as promising candidates due to their ability to convert sunlight and $CO_2$ into valuable products, their minimal growth requirements, and their adaptability to diverse environments. Their metabolic versatility allows for producing a wide range of biofuels, chemicals, and raw materials. Besides biomass [5], the cells can be harvested for a variety of primary and secondary metabolites, such as sugars and alcohols [6, 7], chlorophyll and carotenoids [4], (poly)peptides and human vitamins [8], and terpenoids [9]. In particular, strains of the model cyanobacteria *Synechocystis* sp. PCC 6803 and *Synechococcus elongatus* PCC 7942, are highly attractive platform organisms for the phototrophic production of e.g. isoprene, squalene, valencene, cycloartenol, lupeol or bisabolene [9]. Leveraging the cells' natural capabilities, isolation of molecular hydrogen [10] and reduced nitrogen [4] is also possible, with uses in energy and agronomic sectors. Furthermore, there have been attempts to use cyanobacteria for bioelectricity production [7, 11] or, inversely, to overcome cellular limitations by fuelling cyanobacteria with induced electrical currents [12].

Modifying metabolism for biotechnological purposes involves overcoming natural regulations and inhibitory mechanisms, disrupting the metabolic network's balance. However, balance is crucial for a proper photosynthetic function [13] and, thus, the viability of cyanobacteria for biotechnology. Therefore, a comprehensive understanding of primary and secondary metabolism is essential for effective and compatible modifications. Mathematical models integrate and condense current knowledge to identify significant parts and interactions, enabling the simulation of the effect of various external factors and internal modifications [14, 15]. They can also provide a platform to test new hypotheses. Numerous plant models of primary metabolism helped to identify the most favorable environmental conditions, nutrient compositions, and genetic modifications to maximize the desired outputs [15, 16]. Despite the evolutionary connection between cyanobacteria and plants, the structural and kinetic differences between cyanobacteria and plants (e.g., competition for electrons due to

respiration [17], phycobilisomes (PBSs) as cyanobacterial light-harvesting antennae, photo-protection mediated by Orange Carotenoid Protein (OCP), existence of Carbon Concentrating Mechanism (CCM)) prevent the use of established plant-based models for photosynthesis [3, 17–21]. Even standard experimental methods developed for plants for non-invasive probing of photosynthesis using spectrometric techniques, such as Pulse Amplitude Modulation (PAM) fluorometry and the Saturation Pulse method (PAM-SP) [22], may require either adaptation or change in the interpretation of the measurements when applied to cyanobacteria [3, 23, 24]. In PAM fluorometry, a modulated light source is used to excite the chlorophyll molecules [22]. The emitted fluorescence is then measured, and various parameters derived from this fluorescence signal can provide insights into the efficiency of photosynthesis, the health of the photosynthetic apparatus, and other aspects of plant physiology. Compared to plants and green algae, the measured fluorescence of cyanobacteria has contributions from Photosystem II (PSII), Photosystem I (PSI), and detached Phycobilisome (PBS), resulting in distinct fluorescence behavior [3, 24–26].

Therefore, a mathematical model targeted specifically for cyanobacteria, and capable of simulating and interpreting their re-emitted fluorescence signal after various light modulations is needed to obtain a system perspective on their photosynthetic dynamics. Established cyanobacterial models often describe broad ecosystem behavior or specific cellular characteristics [27]. Worth mentioning here are constrained-based reconstructions of primary metabolic networks [28–30], as well as kinetic models, ranging from simple models of non-photochemical quenching [31] and fluorescence [26] to adapted plant models to study the dynamics of cyanobacterial photosynthesis [32] and models created to study proteome allocation as a function of growth rate [33]. However, none of these models provide a detailed, mechanistic description of oxygenic photosynthesis in *Synechocystis* sp. PCC 6803, including the dynamics of respiration and a mechanistic description of short-term acclimation mechanisms, which are highly sensitive to changes in light wavelengths.

With this work, we provide a detailed description of photosynthetic electron flow in cyanobacteria (as summarized in Fig 1), parameterized to experimental data, including measurements collected under monochromatic light, in *Synechocystis* sp. PCC 6803, a unicellular freshwater cyanobacterium. Light is a critical resource for photosynthetic prokaryotes, which defines their ecological niche and heavily affects cell physiology [24, 34, 35]. Significantly, beyond its intensity, the light spectrum plays a crucial role in exerting physiological control. For example, growth under various monochromatic light sources led to large differences in cyanobacterial growth rate, cell composition, and photosynthetic parameters [36]. Blue light strongly inhibits growth and can cause cell damage by disrupting the excitation balance of photosystems [29, 37, 38], resulting from the varying absorption properties of their pigments [38]. To react to changes in illumination, the cell is able to undergo both short and long-term adaptations. Over time, cells adjust their pigment content (in a process called chromatic acclimation), and the ratio of photosystems to optimize performance [24, 35]. In the short term, processes like OCP-related Non-Photochemical Quenching (NPQ) [39, 40] or state transitions [41] help them adapt, though precise mechanisms of the latter are not yet fully elucidated [3, 42]. While the scientific community agrees that the Plastoquinone (PQ) redox state triggers state transitions, multiple underlying mechanisms have been proposed without a current consensus. Therefore, we also implemented and tested the proposed state transition mechanisms in a model ensemble approach. We found that the PBS-detachment model offered little physiological benefit while PBS movement was the most versatile and robust against high light. Our model uses both light intensity and light wavelengths as input, allowing the simulation of any combination of light sources and adaptation to the specific growth conditions. Readouts include the intermediate metabolites and carriers shown in Fig 1, most importantly Adenosine

**Fig 1. Computational model of the photosynthetic and respiratory chain allows simulating electron fluxes through main (linear), cyclic, alternate, and respiratory pathways in *Synechocystis* sp. PCC 6803.** Schematic representation of components and reactions included in the model of cyanobacterial photosynthesis. The model includes descriptions of protein complexes (e.g. PSII, PSI, C$b_6f$ and ATP synthase) and electron carriers in the photosynthetic electron transport chain and the reactions through them, enabling simulation of electron transfer through Linear Electron Transport (LET), Cyclic Electron Transport (CET), Respiratory Electron Transport (RET), and Alternate Electron Transport (AEF). With orange circles (Respiration, CBB, and photorespiratory salvage pathway (PR salvage)) we mark pathways represented in the model as lumped reactions. The top-right box shows gas exchange reactions ($O_2$ export and active $CO_2$ import) and metabolic ATP and reduced Nicotinamide adenine dinucleotide (NADH) consumption. Electron and proton flows are colored black and blue, respectively. Regulatory effects, such as Fd-dependent CBB activity, are represented with dotted lines. The two photosystems are described using Quasi-Steady-State (QSS) approximation. For the analyzes we assume internal quencher as the state transition mechanism, as marked on PSII. Various scenarios of PBS attachment can be simulated, on the figure attached to PSII. Abbreviations: 2PG: 2-phosphoglycolate, 3PGA: 3-phosphoglycerate, ADP: Adenosine diphosphate, ATP: Adenosine triphosphate, ATPsynth: ATP synthase, CBB: Calvin-Benson-Bassham cycle, CCM: Carbon Concentrating Mechanism, COX: Cytochrome c oxidase, Cyd: Cytochrome bd quinol oxidase, Cyt b$_6$f: Cytochrome $b_6f$ complex, FNR: Ferredoxin-NADP$^+$ Reductase, Fd: Ferredoxin, Flv 1/3: Flavodiiron protein dimer 1/3, NADP$^+$: Nicotinamide adenine dinucleotide phosphate, NADPH: reduced Nicotinamide adenine dinucleotide phosphate, NAD$^+$: Nicotinamide adenine dinucleotide, NADH: reduced Nicotinamide adenine dinucleotide, NDH-1: NAD(P)H Dehydrogenase-like complex 1, NDH-2: NAD(P)H Dehydrogenase complex 2, OCP: Orange Carotenoid Protein, Oxy: RuBisCO oxygenation, PC: Plastocyanin, PQ: Plastoquinone, PR: Photorespiration, PSI: Photosystem I, PSII: Photosystem II, SDH: Succinate dehydrogenase.

triphosphate (ATP) and reduced Nicotinamide adenine dinucleotide phosphate (NADPH), fluxes through several electron pathways: Linear Electron Transport (LET), Respiratory Electron Transport (RET), Cyclic Electron Transport (CET) and Alternate Electron Transport (AEF)), reaction rates, such as carbon fixation and water splitting, and the cell's emitted fluorescence as measured by PAM. We perform Metabolic Control Analysis (MCA) [43–45] of the network in different light conditions, showing that the reactions which determine the rate of Calvin-Benson-Bassham cycle (CBB) flux shift from photosynthetic source reactions to sink reactions within the CBB as light intensity increases. By harnessing the power of mathematical modeling, we seek to provide a computational framework to test further hypotheses on the photosynthetic mechanisms in cyanobacteria and contribute to basic research on these organisms that eventually can lead to optimized cyanobacterial production and contribute to the advancement of green biotechnology.

## Methods

### Model description

We developed a dynamic, mathematical model of photosynthetic electron transport in *Synechocystis* sp. PCC 6803 (further *Synechocystis*) following a classical bottom-up development cycle. Our model consists of a system of 17 coupled Ordinary Differential Equations (ODEs), 24 reaction rates, and 98 parameters, including measured midpoint potentials, compound

concentrations, absorption spectra, and physical constants (Table A in S1 Appendix). By integrating the system over time, we can simulate the dynamic behavior of rates and concentrations of all reactions and reactants visualized on Fig 1 and summarized in Table B in S1 Appendix, including dynamic changes in the lumenal and cytoplasmic pH. We included a detailed description of four commonly distinguished electron transport pathways: LET, CET, RET, and AEF. Given the high similarity between the essential electron transport chain proteins of plants and cyanobacteria [3, 20], the photosystems were described using Quasi-Steady-State (QSS) approximation, as derived in our previous dynamic models of photosynthetic organisms [50, 51]. We followed a reductionist approach simplifying many downstream processes into lumped reactions. The lumped CBB, Photorespiration, and metabolic consumption reactions represented the main cellular energy sinks. Functions describing the CBB and Ribulose-1,5-bisphosphate Carboxylase-Oxygenase (RuBisCO) oxygenation (Oxy) contained multiple regulatory terms, including gas and metabolite concentrations (see S1 Appendix). Although cyanobacteria CCM components include at least four modes of active inorganic carbon uptake [52], we have decided to represent the mechanism as a one lump reaction. By calculating the dissolved $CO_2$ concentration at the cellular pH and with an actively 1000-fold increased intracellular $CO_2$ gas pressure (see Section S1.12 in S1 Appendix) we reflect the very efficient cyanobacterial concentrating mechanism. Unless stated otherwise, simulations were run under three assumptions: 25°C temperature with 230 µM dissolved $O_2$ and supplemented with 5% $CO_2$. The pigment content and photosystems ratio were parametrized to a cell grown under ambient air with 25 µmol(photons) $m^{-2} s^{-1}$ of 633 nm light. All rates and concentrations have been normalized to the chlorophyll content (4 mM). The default initial metabolite concentrations were set to literature measurements (Table C in S1 Appendix). Steady-state simulations were run for $1 \times 10^6$ s. For the steady-state simulations, we considered that the steady-state is reached if the Euclidean norm of relative concentration changes between the last two time steps did not exceed $1 \times 10^{-6}$. Because the regulatory processes, CBB redox activation, OCP activation, and state transitions, have slow rates of change, we compared their relative changes to a threshold of $1 \times 10^{-8}$.

## Code implementation

The model has been developed in `Python` [53] using the modeling package `modelbase` [54] further exploring a highly modular approach to programming mathematical models. The model and scripts used to numerically integrate the system and to produce all figures from this manuscript, as well as analysis run during the peer review process are accessible at https://github.com/Computational-Biology-Aachen/synechocystis-photosynthesis-2024.

## Model parameterization

The model has been parameterized, integrating physiological data and dynamic observations from numerous groups (e.g. pH-ranges: [55], NADPH reduction: [56], $O_2$ change rates: [25, 49] (Fig 2H), $CO_2$ consumption: [33] (Fig 2G), PQ reduction: [57], Plastocyanin (PC), PSI, and Ferredoxin (Fd) redox-states: [58], PAM-SP fluorescence: [42, 59] (Fig 3A), electron fluxes: [46] (WT in Fig 2A and 2E).

The model depends on 98 parameters (Table A in S1 Appendix). 43 parameters, including pigment absorption spectra, were taken directly from the literature, six parameters describe the experimental setup (light: intensity and spectrum, $CO_2$, $O_2$, concentration of cells, and temperature), and eight parameters describing photosystems concentrations and pigment composition were estimated from provided data [59]. The latter parameters were measured spectrophotometrically and through 77K fluorescence, assuming a 10-times higher

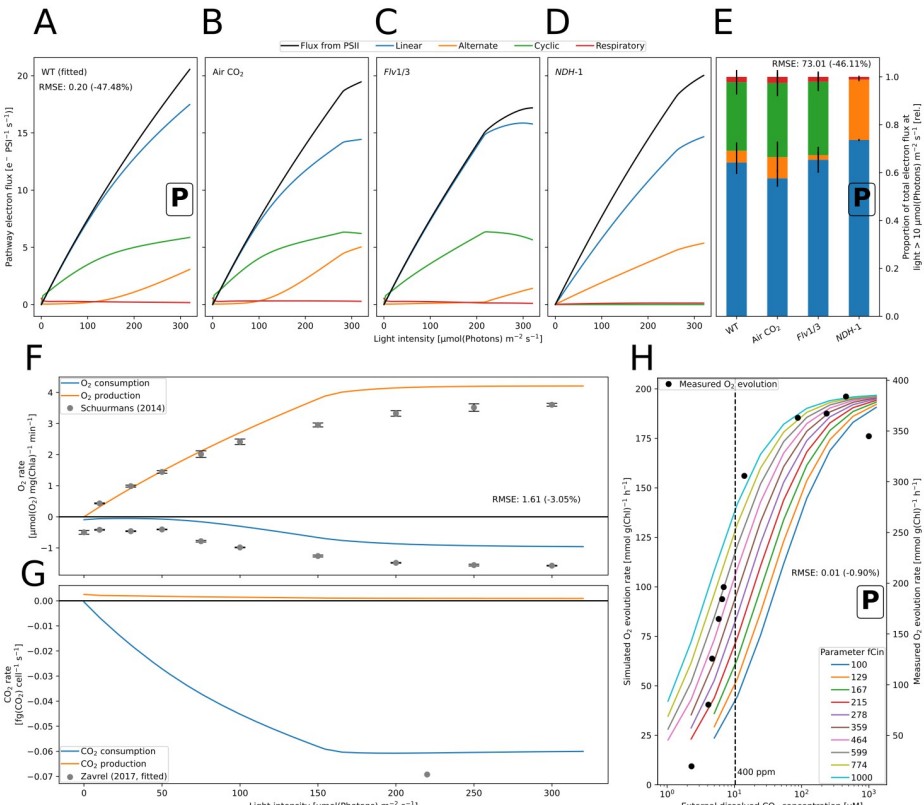

**Fig 2. Simulation of the electron and gas fluxes in *Synechocystis*.** A-D: Simulated steady-state electron flux through linear (blue), cyclic (green), alternate (flavodiiron+terminal oxidases, orange) and respiratory (red) electron pathways for light intensities between 10 μmol(photons) m$^{-2}$ s$^{-1}$ and 300 μmol(photons) m$^{-2}$ s$^{-1}$. The model has been **parameterized** to yield approximately 15 electrons PSI$^{-1}$ s$^{-1}$ linear electron flow (blue) for a fraction of 65% under saturating $CO_2$ conditions, as measured in wild type (WT, A) [46]. The model **predicts** flux distributions under sub-saturating air $CO_2$ level (B) and for the flavodiiron (*Flv1/3*, C) and NAD(P)H Dehydrogenase-like complex 1 (NDH-1) knockout mutants (D). Each value represents a steady-state flux under continuous light exposure. Simulations were run using 670 nm light (Gaussian LED, $\sigma$ = 10 nm). E: Barplot showing the mean flux distribution for light intensities over 10 μmol(photons) m$^{-2}$ s$^{-1}$ ± sd). F: Simulation of oxygen production and consumption (respiration + three terminal oxidases) rates for increasing light intensities, as compared to measured rates provided by Schuurmans *et al.* (2014) [47]. Data points are the mean from measurements of cells with 625 nm illumination with 50 mM NaHCO$_3$. Error bars show ± sd. G: The simulated carbon fixation rates are displayed with the measurement used for parameterization [48]. Simulations were run using a 625 nm light (gaussian LED, $\sigma$ = 10 nm). We calculated light attenuation in the culture using Eq (S67) in S1 Appendix with default pigment concentrations and a constant 2 mg L$^{-1}$ sample chlorophyll concentration. H: $O_2$ production under variation of the external $CO_2$ concentration *in vivo* and *in vitro*. The data was used for parametrization by fitting the parameter *fCin*, representing the ratio of intracellular to extracellular $CO_2$ partial pressure which is increased by the carbon concentrating mechanism, which was varied in the simulation between 100 and 1000. Benschop *et al.* [49] measured oxygen evolution with 800 μmol(photons) m$^{-2}$ s$^{-1}$ light of *Synechocystis* sp. PCC 6803 grown under 20 ppm $CO_2$ and varied the dissolved $CO_2$ in the medium ($C_i$). The data was extracted from graphs using https://www.graphreader.com/. The simulation used a cool white LED. Our simulated $O_2$ evolution rates are ca. half of the measured rates. Within the $CO_2$ concentration range above ambient air (400 ppm) the rate dynamics are well reproduced with *fCin* = 1000. The model overestimates oxygen evolution at very low $C_i$ concentrations. A black box with the letter "P" marks the data used for parameterization. RMSE quantifies the residuals of the respective simulation. In A and E, the residuals measure the difference to an experimentally measured LET flux of 15 electrons PSI$^{-1}$ s$^{-1}$ and 65% of the total PSI electron flux in the WT. The difference to the residuals of a model with initial parameters, not improved with the Monte Carlo results, is in parentheses.

fluorescence yield of free PBS (compare [26]). PAM-SP fluorescence curves were used to fit seven fluorescence-related parameters, including quenching and OCP constants [59], and two parameters were fit to electron transport rate measurements [46]. Nine rate parameters were estimated from reported rates of the reaction itself or connected processes such as $O_2$

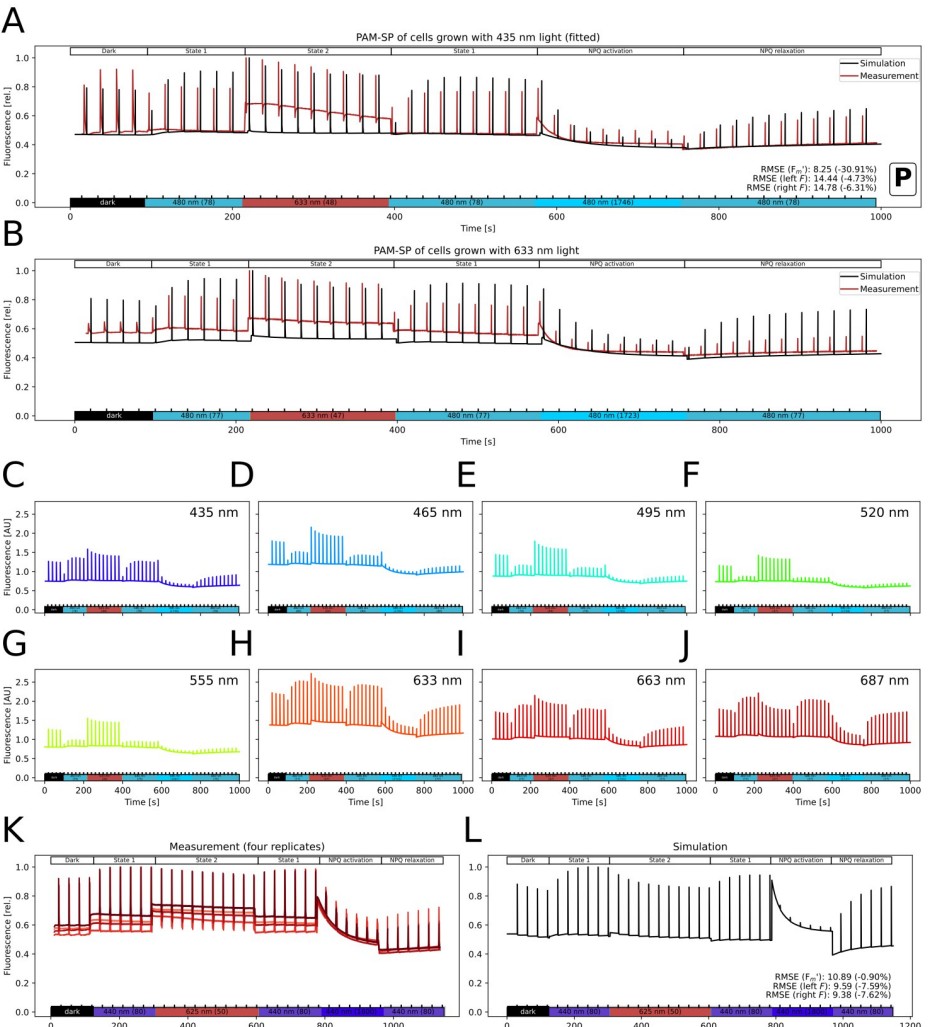

**Fig 3. Saturation pulse method using Pulse Amplitude Modulation (PAM) fluorescence measurement *in vivo* and *in silico*.** The simulated signal has been calculated using Eq (8). All experimental measurements were performed with a Multi-Color PAM (Walz, Effeltrich, Germany). A, B: Fit of simulated (black) to measured (red) PAM fluorescence dynamics during a saturation pulse light protocol. The simulation was manually fit the traces in A and improved by automated parameter variation using seven model parameters, and the parameters are used for all other model simulations. The experimental traces were measured in *Synechocystis* sp. PCC 6803 grown under 435 nm light (A, n = 2) or 633 nm light (B, n = 2) [59]. Simulations use the respectively measured pigment contents and ambient $CO_2$ (400 ppm). The shown light protocol includes several different light wavelengths and intensities to trigger a response from respective photosynthetic electron transport chain components. By monitoring cell responses to these light conditions, we captured light responses via state transitions and non-photochemical quenching and relaxation (as described in the upper bar). We calculated light attenuation in the culture using Eq (S67) in S1 Appendix with the measured pigment concentrations and sample chlorophyll content in Table D in S1 Appendix. C-J: Model prediction on emitted fluorescence signal. Light protocol in A is repeated with pigment contents measured in cells grown under different monochromatic lights [59]. We calculated light attenuation with the chlorophyll content measured in each culture. K,L: Model validation comparing PAM-SP fluorescence traces *in vivo* (K) and *in silico* (L). Four experimental replicates are shown. Simulations assume 1% $CO_2$ supplementation and use the default parameters (see A) and pigment set. The model reproduces the qualitative fluorescence dynamics during most of the experiment except for overestimating steady-state fluorescence during the strong blue light phase. The lower bar depicts the light wavelength and intensity (in parentheses, in μmol(photons) m$^{-2}$ s$^{-1}$) (lights used: 440 nm at 80 μmol(photons) m$^{-2}$ s$^{-1}$ and 1800 μmol(photons) m$^{-2}$ s$^{-1}$ and 625 nm at 50 μmol(photons) m$^{-2}$ s$^{-1}$, saturating pulse: 600 ms cool white LED at 15 000 μmol(photons) m$^{-2}$ s$^{-1}$). Cultures of *Synechocystis* sp. PCC 6803 were grown under bubbling with 1% $CO_2$ and 25 μmol(photons) m$^{-2}$ s$^{-1}$ of 615 nm light for ca. 24 h. For the measurement, 1.5 mL culture was transferred to a quartz cuvette and dark-acclimated for 15 min prior to each measurement. A black box with the letter "P" marks the data used for parameterization. RMSE quantifies the residuals of the respective simulation. The difference to the residuals of a model with initial parameters, not improved with the Monte Carlo results, is in parentheses.

generation for respiration. To derive rate constants, we divided the determined rate by the assumed cellular substrate concentrations. Five parameters stemmed from simplifying assumptions regarding inhibition constants, the cytoplasmic salinity, and pH buffering. 16 further parameters were fitted to reproduce literature behavior such as cellular redox states or regulation of the CBB. The weighting factors of PSI and PSII fluorescence were initially set to one.

We have initially parametrized the model manually by fitting model simulations to the previously stated data. The robustness of this parametrization was tested using Monte Carlo simulations. We simulated the model with many randomized parameter sets, testing if any set produced simulations with lower residuals to all datasets the model was compared to. If no change to the parameters can improve all of these objectives at once, the parameter set is considered to be on the Pareto front [111]. Finally, the model was improved by using the best parameter set found in the Monte Carlo simulations. During model refinement, we checked modifications to the model quantitatively by calculating residuals to experimental data and visually for changes in trends of our simulated redox state, oxygen evolution, carbon fixation, and dynamics of implemented photoprotective mechanisms. A comprehensive list of all model parameters utilized in this study, including values needed for unit conversion, is provided in Tables A and D in S1 Appendix (state transition analysis separate, see below), ensuring transparency and reproducibility of our computational approach.

## Reaction kinetics

Following the principle of parsimony, all reactions where no additional regulatory mechanism was known have been implemented with first-order Mass-Action (MA) kinetics. A reaction with substrates $S_i$ and products $P_j$ is defined as: $\Sigma n_i S_i \leftrightarrow \Sigma m_j P_j$ with $i, j \in \mathbb{N}$ where $n_i$ and $m_j$ are the stoichiometric coefficients of substrates and products, respectively. For each reaction, we calculated the Gibbs free energy ($\Delta_r G'$, see supplemental information) [50, 51]. Only reactions with $\Delta_r G'$ close to 0 under physiological conditions were described with reversible kinetics. Thus, we set reactions as irreversible except for ATP synthase, Succinate Dehydrogenase (SDH), Ferredoxin-NADP$^+$ Reductase (FNR), regulatory variables (e.g. CBB activation), and PSII and PSI internal processes.

To simplify higher order reversible MA equations [60], we first decompose the rate equation into separate kinetic and thermodynamic components (as done for Michaelis-Menten (MM) [61]) and then simplify only the kinetic part, leading to (see Eqs (S3) and (S5) in S1 Appendix):

$$
v = \begin{cases}
\text{if } \Delta_r G' < 0: & \overbrace{k^+ \cdot \prod c_{S_i}}^{\text{kinetic}} \cdot \overbrace{(1 - \exp(\Delta_r G'/RT))}^{\text{thermodynamic}} \\[2em]
\text{else}: & k^- \cdot \dfrac{\prod c_{P_j}}{K_{eq}} \cdot (\exp(-\Delta_r G'/RT) - 1)
\end{cases}
\tag{1}
$$

with substrate concentrations $c_S$, product concentrations $c_P$, and $K_{eq} = \exp(-\Delta_r G'^0/RT)$. Here, we approximate $k \cdot \prod c_{S_i}^{n_i} \approx k^+ \cdot \prod c_{S_i}^{n_i} / \prod c_{S_i}^{n_i-1}$ and $k \cdot \prod c_{P_j}^{m_j} \approx k^- \cdot \prod c_{P_j}^{m_j} / \prod c_{P_j}^{m_j-1}$ which, for any $n_i > 1$ or $m_j > 1$, leads to $k^+ \neq k^-$ (see Eq (S6) in S1 Appendix). This necessitated parameterising $k^+$ and $k^-$ separately. The reactions FNR and SDH, which were deemed reversible, used rate Eq (1) with the determination of $\Delta_r G'$ during simulation. We calculated electron pathway fluxes from the following involved reactions: LET (FNR), CET (NAD(P)H Dehydrogenase-like complex 1 (NDH-1)), respiration (SDH, NAD(P)H Dehydrogenase complex 2 (NDH-2)), and AEF (Cyd, COX, Flavodiiron protein dimer 1/3 (Flv)) (see Eq (S47) in S1 Appendix).

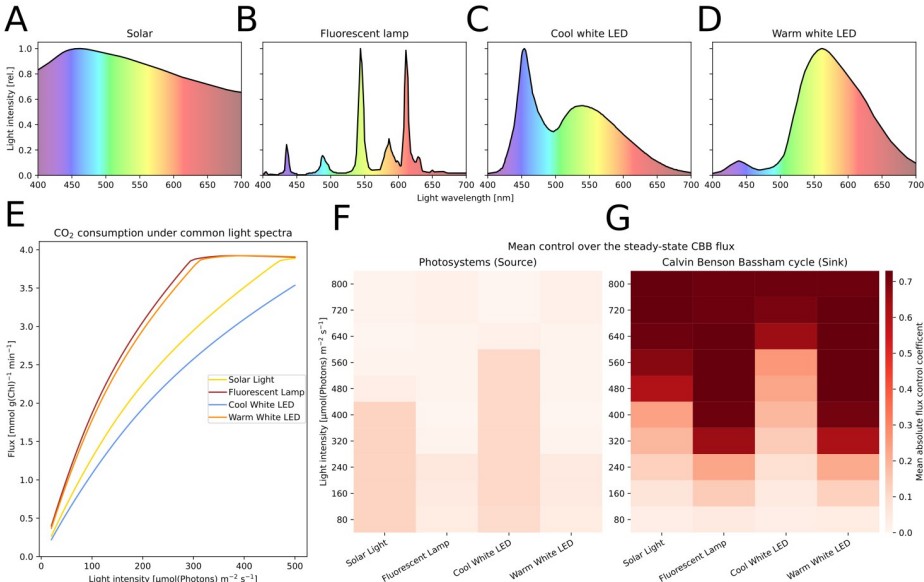

**Fig 4. Systematic analysis of the effect of various light sources on the rate of carbon fixation.** A-D: The light spectra of four light sources used in simulations, including common "white" light LED panels. E: Steady-state flux through the CBB under the lights in A-D, simulated over different light intensities. F,G: Results of Metabolic Control Analysis (MCA) performed for the given light sources with a range of intensities from 80 to 800 µmol(photons) m$^{-2}$ s$^{-1}$ to steady-state. By varying the photosystem concentrations and the maximal rate of the CBB by ± 1%, we quantified their control on the CBB flux by calculating flux control coefficients. We display the mean of absolutes of control coefficients. Higher values signify stronger pathway control.

## Implementation of monochromatic and polychromatic light sources

To consider the influence of light spectrum on photosynthetic activity, our model takes light as input ($I$) with wavelengths ($\lambda$) in range between 400 and 700 nm. In this work we performed simulations using the solar spectrum, a fluorescent lamp, cold white LED, warm white LED, and "gaussian LEDs" simulated as Gauss curves with a set peak-wavelength and variance of 10 nm or 0.001 nm ("near monochromatic") [62] (see Figs 4A–4D and 5A).

For calculation of absorbed light we further differentiate between the light absorbed by PSI, PSII, and PBS, based on their reported pigment composition [62]. We focused on four most abundant pigments: chlorophyll, β-carotene, phycocyanin, and allophycocyanin, although the implementation allows for more complex composition.

We assume that PBSs can be either free, in which case the excitation is lost, or attached to one of the photosystems to transfer their excitation energy. The respective fractions of PBS states were fixed to values from [59], except for simulations of state transition mechanisms which required dynamic PBS behavior. We assumed pigment content and PBS-attachment as measured by Zavřel *et al.* [59], although different pigment composition can be provided as an input to the model. We calculate PSII excitation rate $E_{PSII}$ (and $E_{PSI}$ analogically) as:

$$E_{PSII} = P^{PSII} \cdot Q_{OCP} \cdot \text{simpson}(A \cdot \text{diag}(I)) \cdot lcf \tag{2}$$

where vector $P_{1\times4}$ describes how excitations from the four pigments are distributed onto the photosystems, accounting e.g. for their pigment composition, high PSI:PSII ratio, the PBS attachment, and spillover; $Q_{OCP} = \text{diag}(1, 1, 1 - OCP, 1 - OCP)$ is a diagonal matrix with values set to one everywhere but at the contribution of PBS to reduce the excitation rate by light energy quenched due to OCP activity; $A_{p\times\lambda} = (a_{\lambda,p})$ contains each pigment $p$'s-specific

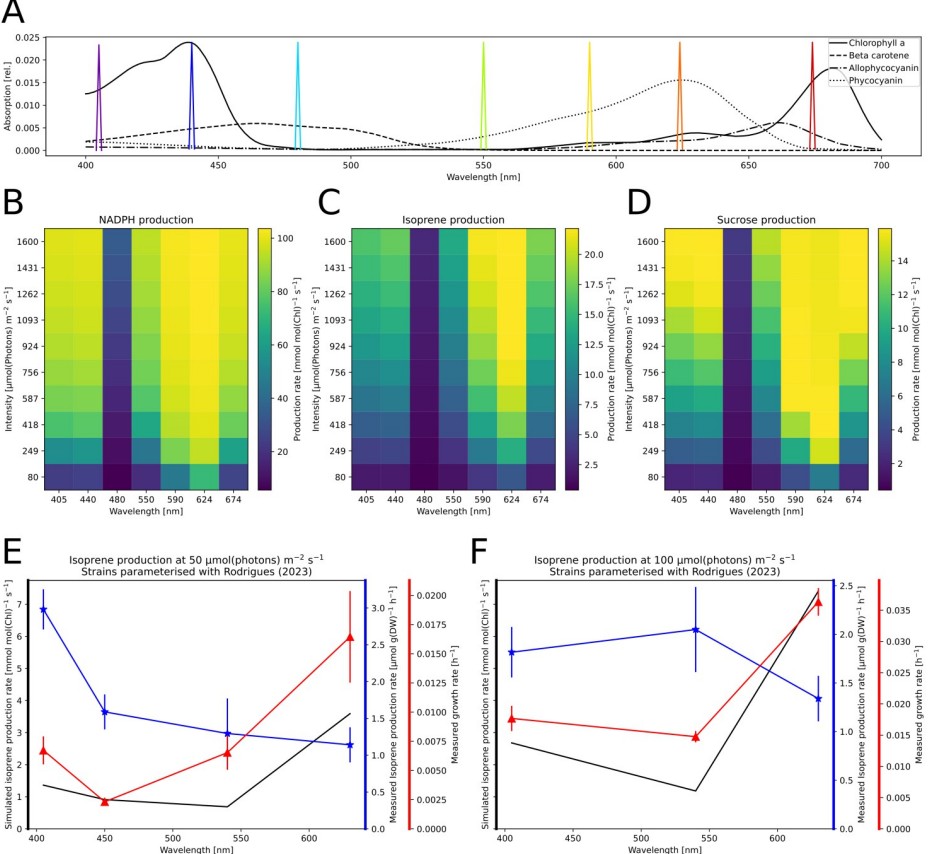

**Fig 5.** *In silico* **analysis of biotechnological compound production.** A: Monochromatic lights used in the analysis (Gaussian LED, $\sigma = 0.001$) shown as colored spikes. Relative absorption spectra of selected pigments are shown in the background. B-D: Simulated production capacities of biotechnological compounds under light variation. We created three models, each containing a sink reaction consuming energy carriers in the ratio corresponding to a biotechnological target compound, including the cost of carbon fixation. The models were simulated to steady-state under illumination with varying intensity of the lights in A. We disabled the CBB reaction and limited ATP and NADPH concentrations to 95% of their total pools. Thus we estimate the maximal production rate of energy carriers in a desired ratio, assuming optimal carbon assimilation for the process and no product inhibition. The shown sinks represent pure NADPH extraction (B), production of terpenes (C; 19 ATP, 11 NADPH, 4 Fd$_{red}$), and sucrose (D; 19 ATP, 12 NADPH). E,F: Comparison of measured isoprene production and simulated production capacity under light variation. We show the growth and isoprene production rates measured under monochromatic illuminations (405 nm, 450 nm, 540 nm and 630 nm at 50 (E) or 100 µmol(photons) m$^{-2}$ s$^{-1}$ (F)) by Rodrigues *et al.* [29]. The model was adapted to the measured pigment composition (see Table D in S1 Appendix) and simulated to steady-state under the respective monochromatic lights. We implemented the sink reaction representing isoprene as in C.

absorption spectra; "simpson" is a row-wise, numerical integral of the light absorbed by each considered pigment, calculated using the composite Simpson's rule (we used `scipy.integrate.simpson` function); *lcf* = 0.5 is the light conversion factor to estimate the amount of generated excitations, which was fitted to match the wild-type electron transfer rates of Theune *et al.* [46] (Fig 2A). Importantly, we assume that despite the wavelength-dependent energy content, all photons result in equivalent excitation of photosystems with the extra energy being lost as heat [63, 64]. This implementation enabled us to simulate various light-adapted cells by updating the parameters corresponding to measured pigment composition and photosystem ratio. For simulations of PAM-SP, we further calculate the light encountered by a mean cell (*I*) for each wavelength according to an integrated Lambert-Beer function [65]

accounting for the decreasing irradiance at various depths due to cellular absorption (see Eq (S67) in S1 Appendix).

## Activation of photosystems

Following our previous approach [50], we modeled the photosystem's excitation and internal electron transport assuming that i) they are at Quasi-Steady-State (QSS), as they operate on a much faster time scale than other photosynthetic reactions, and ii) at every time point, photosystem II can be in one of four possible states, and PSI in three, relating pigment excitation with the charge separations at reaction centers (S1 Fig). The PSII excitation rate constant $k_{LII}$ is calculated from $E_{PSII}$ in Eq (2) (in µmol(photons) mg(chl)$^{-1}$) multiplied by the molar mass of chlorophyll $M_{Chl}$ and normalized to PSII concentration ($c_{II}$):

$$k_{LII} = E_{PSII} \cdot M_{Chl} \cdot \frac{1}{c_{II}} \tag{3}$$

PSII was described with four ($B_0$—$B_3$) and PSI with three states ($Y_0$—$Y_2$) (see Fig 1 and S1 Fig). The QSS models also consider the relaxation of excitations by fluorescence or heat emission (only PSII). We defined the PSII rate $v_{PSII}$ as

$$v_{PSII} = 0.5 \cdot k_2 \cdot B_1 \tag{4}$$

since two $B_1 \rightarrow B_2$ reactions have to occur for a full PQ reduction. $k_2$ is the rate constant of the photochemical quenching.

## Calculating the fluorescence signal

Based on the principle of PAM measurements, the model calculates fluorescence proportional to the gain in excited internal states of PSII and PSI when adding measuring light to the growth light. Additionally, we consider fluorescence of free PBS using their light absorption [26]. The default measuring light is set to 625 nm at 1 µmol(photons) m$^{-2}$ s$^{-1}$ throughout this manuscript.

PAM fluorometry measures the cellular fluorescence emitted in response to microsecond pulses of measuring light with a constant, low intensity. We built our model fluorescence function on the same principle. Measuring light pulses with irradiance $I_{ML}$ are applied on top of the actinic light $I$, so cells experience a total irradiance of $I_{+ML} = I + I_{ML}$. We then recalculate the photosystems' QSS systems using $I_{+ML}$, resulting in the internal states $B_0^{+ML}$ to $B_3^{+ML}$ and $Y_0^{+ML}$ to $Y_2^{+ML}$. We then define the PAM fluorescence as the increase in photosystem fluorescence reactions by the addition of measuring light (see S1 Fig):

$$F_{PSII} = k_F \cdot \left(B_1^{+ML} - B_1 + B_3^{+ML} - B_3\right) \tag{5}$$

$$F_{PSI} = k_{F1} \cdot \left(Y_1^{+ML} - Y_1\right) \tag{6}$$

where $k_F$ and $k_{F1}$ are the rate constants of flurescence emissions by PSII and PSI, respectively. Lastly, we make the simplifying assumption that PBS fluorescence only results from the fraction of uncoupled PBS $f_{free}^{PBS}$ and is proportional to their absorption of $I_{+ML}$:

$$F_{PBS} = \text{simpson}_\lambda\left(p_{\lambda,PBS} \cdot \text{diag}\left(I_{+ML}\right)\right) \cdot f_{free}^{PBS} \tag{7}$$

where $p_{\lambda,PBS}$ is a vector of the PBS absorption spectrum. Considering cyanobacterial optical properties of light-harvesting pigments, fluorescence measured at room temperature can originate from both photosystems and PBS [3]. There have been attempts to determine the

fluorescence contributions of each component [26]. We assume that the three fluorescent species contribute differently to the fluorescence detected > 650 nm, e.g. because of differing emission spectra. Therefore, we include weighing factors when calculating the total recorded fluorescence in Eq (8), which were calculated in Fig 3A fitting. We estimate the total measured fluorescence signal by calculating the weighted sum of PSI, PSII, and PBS fluorescence:

$$F = w_{PS1} \cdot F_{PS1} + w_{PS2} \cdot F_{PS2} + w_{PBS} \cdot F_{PBS}, \tag{8}$$

where weights $w$ were manually fitted to reproduce the fluorescence dynamics under changing light conditions of the experiment displayed in Fig 3 (fitted values can be found in Table A in S1 Appendix as `fluo_influence`).

## Testing four possible mechanisms of state transitions

We use the model to provide arguments for a possible mechanism of state transitions that is not yet fully elucidated. We have implemented and tested four proposed state transition mechanisms based on a recent review [42] (Fig 6A). We model the transition to state 2 depending on reduced PQ ($PQ_{red}$) and to state 1 on oxidized PQ ($PQ_{ox}$). We implemented the default PSII-quenching (used for simulations in Fig 3A) using a constitutively active quenching reaction and a reverse reaction being activated by $PQ_{red}$ following a Hill equation. The remaining state transition models were described with few reactions and using MA kinetics. A complete

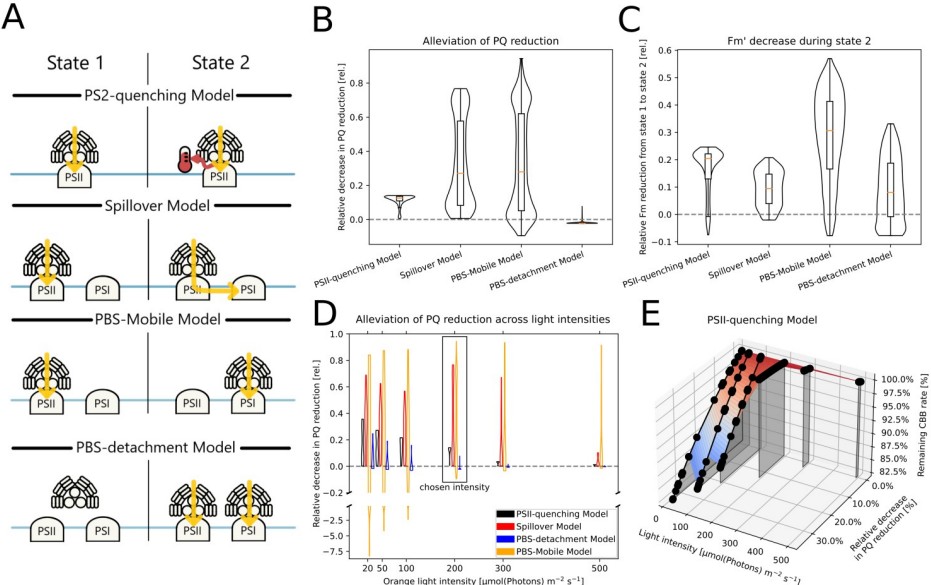

**Fig 6. Testing possible mechanisms of state transitions.** Four possible mechanisms of state transitions have been implemented and parameterized randomly to quantify their performance by contributing to oxidising PQ pool. The fluorescence signal has been calculated using Eq (8). A: Schematic representation of four state transition models, as reviewed in [42]. B: The relative difference of PQ reduction under state 2 lighting to a model with no state transitions. The respective implementations' parameters were systematically varied, and the distribution of relative reduction-alleviation is shown as violin plots with quantiles as boxplots. C: The reduction of maximal fluorescence in the light ($F'_m$) from state 1 to state 2 in the model variations. The distribution is also shown as violin plots with inset boxplots. D: The relative decrease in the reduction state of PQ was calculated for different intensities of 633 nm light. The distribution is shown for each combination of state transition model and light intensity. Dots mark the median. E: Results of the systematic parameter variation of the PSII-quenching model. Each point shows a model with varied parameters for the state transition model. The models were simulated for different light intensities on the x-axis, and we calculated how the activation of the state transition mechanism affected the reduction of the PQ pool (y-axis) and the rate of the CBB (z-axis). For higher light intensities, the state transition has less effect on the PQ pool and CBB rate.

mathematical description of the implementations is available under paragraph S1.6 in S1 Appendix. For the analysis, we systematically varied the parameter sets of all implementations within approximately two orders of magnitude and compared the steady-state fluorescence and PQ redox state under different lighting conditions (actinic: 440 nm at 80 μmol(photons) m$^{-2}$ s$^{-1}$ or 633 nm at 200 μmol(photons) m$^{-2}$ s$^{-1}$; measuring: 625 nm at 1 μmol(photons) m$^{-2}$ s$^{-1}$). We further repeated the PQ redox state analysis under different light intensities between 20 and 500 μmol(photons) m$^{-2}$ s$^{-1}$ (parameters in Table E in S1 Appendix).

## Metabolic control analysis

Metabolic Control Analysis (MCA) is a quantitative framework to study how the control of metabolic pathways is distributed among individual enzymes or steps within those pathways. It quantifies the change in steady-state compound concentrations or reaction fluxes in response to perturbation of an examined reaction [43, 45]. We used the `modelbase.mca` function `get_response_coefficients_df` to perform MCA on our model. The function is using definitions proposed by [43, 66] and calculates the flux control coefficients ($C_{v_k}^{J_j}$) and concentration control coefficients ($C_{v_k}^{S_j}$) using formulas:

$$C_{v_k}^{J_j} = \frac{v_k}{J_j} \frac{\partial J_j / \partial p_k}{\partial v_k / \partial p_k}, \tag{9}$$

$$C_{v_i}^{S_j} = \frac{v_k}{S_j} \frac{\partial S_j / \partial p_k}{\partial v_k / \partial p_k}, \tag{10}$$

where $J_j$ and $S_j$ are respectively the steady-state fluxes and concentrations of the system, $p_k$ is a kinetic parameter which affects directly only reaction $k$ with the rate $v_k$ (see [43, 66]). We approximated these formulas numerically using the central difference, varying the parameters by ±1%. MCA has been repeated for various simulated irradiances (Fig 4A–4D). For systematic analysis of the effect of various light sources on the rate of carbon fixation, we calculated the absolute of the control coefficients and show the mean of the CBB itself and the following sets of model reactions: Photosystems (PSI, PSII), light-driven (PSI, PSII, Cytochrome $b_6f$ complex ($Cb_6f$), NDH-1, FNR), alternate (Flv, Cytochrome $bd$ quinol oxidase (Cyd), Cytochrome $c$ oxidase), and respiration (lumped respiration, Succinate Dehydrogenase, NDH-2).

## Analysis of the production capacity

Exploring the highly modular structure of the model, for determining the production potential of a biotechnological compound, we added an irreversible model reaction consuming ATP, NADPH, and Fd in the required ratio. We assume optimality of carbon provision by the CBB and, thus, set its rate to zero and add the energy equivalent cost of carbon fixation to the cost of the biotechnological compound. The sink reaction was described using hill kinetics with a vmax sufficient to prevent substrate accumulation under any light intensity (here set to 5000 mmol mol(chl)$^{-1}$ s$^{-1}$ for every sink). We use a Hill coefficient of four for high cooperativity and low ligand concentration producing half occupation to achieve a high sink rate but avoid fully draining the pool of any substrate. We set the ligand concentration producing half occupation for PQ, PC, Fd, NADPH, NADH and ATP to 10% of the metabolites total pool, for 3PGA to 1 mmol mol(chl)$^{-1}$ and for cytoplasmic protons to $1 \times 10^{-3}$ mmol mol(chl)$^{-1}$. Additionally, we added MA reactions draining ATP and NADPH with a very high rate constant (10 000s$^{-1}$) if their pools became filled over 95% to avoid sink limitation by either compound.

## Overexpression analysis

The MCA showed that the flux control of FNR and $Cb_6f$ over the CBB was strongly light dependent. The control coefficients of both reactions differed between 440 and 624 nm lights. To test how this would affect a biotechnological approach, we simulated an overexpression of the reactions by increasing a rate-determining parameter of the respective reaction by a factor of two. We then simulated the steady-state carbon fixation rate under light intensities between 0.1 and 500 μmol(photons) m$^{-2}$ s$^{-1}$ and compared them to a model with default parameters.

## Residuals and multiobjective function

To quantify the deviance between simulation and data, we calculated model residuals as the root mean squared error (RMSE). In total, we defined ten residual functions (R1-R10): We simulated the steady-state electron fluxes under 670 nm light at ten intensities between 100 and 300 μmol (photons)m$^{-2}$ s$^{-1}$. We then calculated the fraction of LET of these fluxes and compared their mean to the experimentally determined ratio of ca. 65% ([46], Fig 2E) (R1). Also, we compared the simulated LET flux at 300 μmol(photons) m$^{-2}$ s$^{-1}$ to the experimentally measured value of ca. 15 electrons PSI$^{-1}$ s$^{-1}$ ([46], Fig 2A) (R2). Next, we simulated steady-state $O_2$ production and consumption at different 625 nm light intensities ([47], Fig 2F) (R3). We also simulated steady-state $O_2$ production under variation of the external $CO_2$ concentration (simulated light: 800 μmol(photons) m$^{-2}$ s$^{-1}$ of cool white LED light) ([49], Fig 2H) (R4). For the PAM-SP fluorescence data, we simulated the whole experiment and normalized both data and simulated fluorescence to the respectively highest recorded value. We then determined the height of maximal fluorescence in the light ($F'_m$) and of steady-state fluorescence (F) 3 s before and 3 s after each $F'_m$. The residuals were calculated separately for the values of $F'_m$, and the two F levels. This lead to three residual values for the datasets in Fig 3A ([59]) (R5-R7) and Fig 3K and 3L (R8-R10).

The time for the computational determination of residuals was limited to ten minutes per model. We set this limit to avoid computations being blocked by models that might not reach steady-state.

## Monte Carlo simulations

For Monte Carlo simulations, we selected the 24 parameters denoted as"manually fitted" in Table A in S1 Appendix, as their values are less certain. The parameter fluo_influence among them is a vector containing three values, meaning 26 parameters were varied in total. We then created 10,000 models where the original, manually fitted values of these 26 parameters were varied randomly within ±factor 2 or ±factor 0.1: Each parameter was multiplied with a random factor $x$ drawn from a log-uniform distribution: $ln(x) \sim U\left(ln\left(\frac{1}{\text{factor}}\right), ln(\text{factor})\right)$ (S2 Fig). The previously described residual functions were calculated for each model. If any simulation failed during the calculation or a calculation timed out, we excluded the respective model from further analysis.

A small fraction of models where parameters were varied by a factor of 0.1 showed improvement of all residuals in the multi-objective function. Therefore, we decided to improve our parameter estimation by using such a parameter set, bringing the model closer to the Pareto front. However, to avoid mixing parametrization and validation data, we chose the parameter set that minimized only the residuals from parametrization data (Figs 2A, 2E, 2H and 3A).

## Parameter optimisation

We additionally used an optimization algorithm to find an improved parameter set using multi-objective optimization, that simultaneously improves all residuals compared to the

default parameter set. We defined the optimization function for a parameter set $p$ as the mean of the normalized residuals:

$$opt(p) = \frac{\sum_{i=1}^{10} R_i(p)/r_i}{10} + F(p) \tag{11}$$

where $r_i$ is a fixed normalization factor that scales each residual of the default model to around one. We also added a penalization function F, that increased the optimization function by 50 if any residuals increased by more than 1% compared to the default model. The `scipy` function `minimize` was used to minimize the optimization function using the Nelder-Mead algorithm.

## Results

We present the first kinetic model of photosynthesis developed for cyanobacteria that can simulate its dynamics for various light intensities and spectra. It is developed based on well-understood principles from physics, chemistry, and physiology, and is used as a framework for systematic analysis of the impact of light on photosynthetic dynamics. Our analysis focuses on several key aspects: the redox state of electron carriers, carbon fixation rates in ambient air, reproduction of fluorescence dynamics under changing light conditions, and the electron flow through main pathways (LET, CET, AEF) under different conditions. We also calculated the fraction of open PSII for increasing light intensities to assess the model quality (S3 Fig). We observed that our response curve is less sensitive to increasing light, as our PSII are open for higher light intensities than reported 300 µmol(photons) m$^{-2}$ s$^{-1}$ [67]. Our model also includes a description of carbon uptake which was parameterized by fitting simulation results to oxygen production measurements by Benschop *et al.* [49]. As the maximum of the reported simulated values differed quantitatively (400 and 200 mmol g(chl)$^{-1}$ h$^{-1}$, respectively) we aimed for a qualitative fit of the dynamics (Fig 2H). Unfortunately, the exact culture conditions (e.g. density) used in the reference work [49] are not known. Hence, factors affecting the estimated oxygen production, such as the pigment composition, may differ, and validation against an independent dataset was necessary. In comparison to measurements by Schuurmans *et al.* [47], we achieved quantitative agreement under low light and exceeded the measured rates under light saturation by only ca. 20% (Fig 2F).

### Robustness analysis

We quantified the deviance between our model and the data employed in this manuscript (Figs 2A, 2E, 2F, 2H, 3A, 3K and 3L). We calculated the Root Mean Squared Error between the simulations and data, leading to 10 residual functions. In the parametrization of the model, some parameters were manually fitted to reproduce data or expected behavior. To test the robustness of this parametrization, we performed Monte Carlo simulations by simulating 10,000 models with parameters randomized by a factor of 2 around their default values. No model among the 9,527 successful simulations improved all residuals at the same time (Fig 7A) with the mean across all relative residuals being 1.51. A second Monte Carlo simulation with parameters varied only within 10% found 439 simulations that improved all residuals (Fig 7B). Therefore, our manual fitting did not result in a model directly at the Pareto front. However, the model is robust against moderate parameter changes. As a result, we improved our model by using the best parameter set from the Monte Carlo analysis from there on.

We used a minimization algorithm to investigate whether a model with improved residuals would considerably improve our analysis. We minimized the mean of all residual functions prohibiting any residuals from increasing compared to the initial model. That way, we avoided fitting the parameter set to only one dataset while sacrificing the fit to other data. The

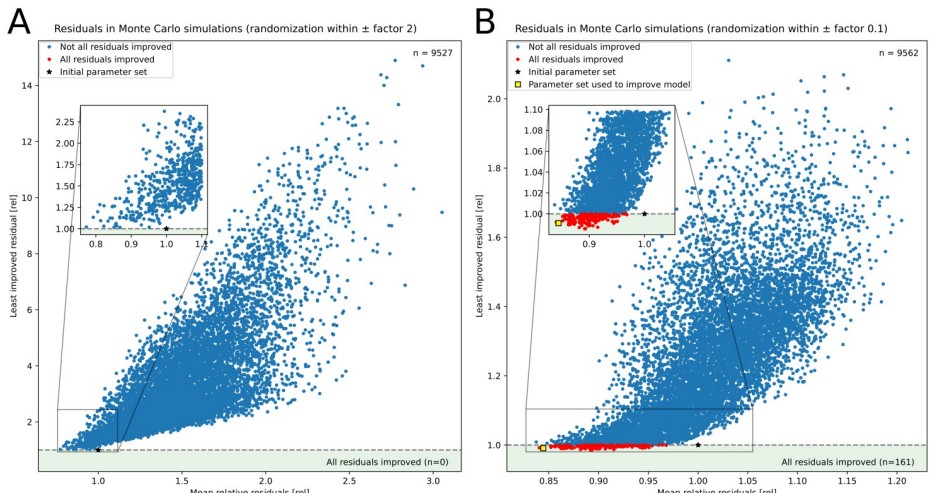

**Fig 7. Residuals of the model in Monte Carlo simulations.** We performed 10,000 simulations of the model with a subset of parameters being randomized: The 24 parameters marked as "manually fitted" in Table A in S1 Appendix were randomly varied within ±factor 2 (A) or ±10% (B) around their initially set value. We calculated ten residual functions for each simulation as described in the Methods. The x-axis shows the mean ratio of residuals compared to the initial parameter set, while the y-axis shows the highest ratio. Simulations with a y value below one, ca. 1.6% of the simulations in B and colored red improve all residual functions. We mark the parameter set with initial values for the randomized parameters and the set whose values were used to improve the model.

optimized parameter set reduced the mean residuals by 17% of the previous parameter set. Upon simulating individual experiments again, no new features were captured, and the improvement was almost negligible. For example, when the simulation of Fig 2A was repeated with the optimized parameters, the residuals of F decreased by 4 to 8% while the residuals of $F'_m$ increased by 12% (S4A Fig). This change was mainly caused by a slower engagement of state transitions and a generally lower $F'_m$ level under OCP activation. However, the optimized parameter set did not change the response to changing light, like decreasing $F'_m$ during the transition to state 2 in red light or the activation of OCP in strong blue light. Similarly, $O_2$ production measured by [47] was matched 3% better, but the simulated rate under high light stayed the same (S4C Fig).

## Flux through alternative electron pathways

We simulated the steady-state flux of electrons through the Photosynthetic Electron Transport Chain (PETC) for four transport pathways under 670 nm monochromatic illumination (Fig 2A–2D). We parameterized the flux through the LET to yield approximately 15 electrons $PSI^{-1}$ $s^{-1}$ and 65% of the total PSI electron flux in the wild type (WT) [46]. Our simulated saturation of CET around 300 μmol(photons) $m^{-2}$ $s^{-1}$ compares well to proton flux measurements by Miller *et al.* [67]. Under ambient $CO_2$ (400 ppm), our model simulates an overall limitation of electron flux and an increase in alternative flows. We found similar electron partitioning between WT and in the *Flv1/3* mutant at lower light intensities agreeing with the findings of Theune *et al.* [46]. However, our simulations show significant AEF in the WT over 200 μmol (photons) $m^{-2}$ $s^{-1}$, which might have been suppressed by high $CO_2$ and pH in the experiments by Theune *et al.* (personal correspondence, see also [68]).

Under intermediate light intensity, the *Flv1/3* mutant also showed a higher CET while maintaining LET similar to the WT, pointing towards a balancing act of NDH-1. Inversely, our simulated NDH-1 mutant maintained high AEF but, in contrast to Theune *et al.*,

significant flux through the LET. In addition to simulating electron flow, our model can probe the intracellular redox state, pH, and additional fluxes through key biochemical reactions (S5 Fig) and simulate the expected results of fluorescence analysis (S6 Fig). For example, it can be seen that a reduced PQ pool under high light leads to reduced CET mediated by NDH-1 and, in turn, a decreased CBB flux due to insufficient provision of ATP. Furthermore, we find that mutations affecting the electron flow lead to an increased Non-Photochemical Quenching (NPQ) at higher light intensities and the decrease in photosynthetic yield (S6 Fig).

## Photosynthesis dynamics captured via fluorescence measurements

Using experimental measurements (pigment concentrations, photosystem ratios, and expected PBS-attachment—method currently under review) [59], we manually fitted model parameters to represent a *Synechocystis* strain grown under 435 nm monochromatic light (Fig 3A). With this model, we simulate fluorescence in a PAM-SP light protocol [22], which investigates photosynthesis behavior using the dark-adapted minimal ($F_0$) and maximal fluorescence ($F_m$), the maximal fluorescence in the light ($F'_m$) and the constantly measured steady-state fluorescence (F). By monitoring cell responses to changing light conditions, we captured light responses via state transitions and non-photochemical quenching and relaxation (for a review of the mechanisms, see [3] and for related models in plants [15]). We simulated the same light protocol of blue and red light, as used *in vivo* [59]. Our simulation qualitatively reproduces the transition between states 1 and 2 and the activation and relaxation of NPQ by the Orange Carotenoid Protein (OCP). Because our model underestimates PSII closure in response to light, the steady-state fluorescence during light phases is also underestimated. By systematically comparing our simulation results and experimental data, we have revealed that the experimentally used saturation pulses were non-saturating in 480 nm actinic light and induced fluorescence quenching, as confirmed by follow-up experiments (see Fig S14 in [59] and S7 Fig in this work). Thus, we found the model's usefulness in investigating fluorescence measurements. Using the same fitted parameters, we can also reproduce the qualitative behavior of cells grown under 633 nm monochromatic light (Fig 3B) and predict the fluorescence under further adapted pigment contents (Fig 3C–3J). The model shows a strong effect on cellular reactions and fluorescence when adapted to pigment contents of cells grown under other monochromatic lights. We have further validated our model against the newly measured fluorescence trace Fig 3K and 3L. Our simulations predict well the dynamics of $F'_m$, but overestimate the fluorescence signal in high 440 nm light (Fig 3K and 3L).

## Common light sources affect the metabolic control differently

Photosynthesis experiments can be conducted with many different light sources that are equivalent in photon output but differ in the spectrum. To further investigate how these spectral differences affect cellular metabolism, we simulated the model with different monochromatic and"white" light sources: solar irradiance, fluorescent lamp, and cool and warm white LED (Fig 4A–4D). For each light, we simulated the model to steady-state to perform MCA (Fig 4F and 4G). We perturbed single parameters of the PETC components by ± 1% and quantified the effect on the steady-state fluxes and concentrations. A high control coefficient represents a strong dependency of the pathway flux on changes to that parameter, with control in a metabolic network being distributed across multiple reactions. A single parameter being in full control of the flux through a network would represent the case of a typical bottleneck, but this rarely occurs in biological systems [43, 45]. We show that the electron pathway-specific control differs between the simulated light sources. Our results indicate that, at lower intensities of solar and cool white LED light, the control mainly lies within the photosystems as sources of

energy carriers (Fig 4F). We find less control by the photosystems for light spectra with a higher proportion of red wavelengths, suggesting such light sources induce less source limitation. Accordingly, the maximal simulated $CO_2$ consumption is reached at lower light intensities for these spectra (Fig 4E). All tested spectra show the CBB having the main control of $CO_2$ fixation only under increased light, marking a shift towards the energy carrier sink limitation.

Repeating the analysis with simulated monochromatic lights, we found similar differences that seem to correspond with the preferential absorption by either chlorophyll or PBS (S8 Fig). The earliest switch to sink limitation was found in 624 nm light, while light that is weakly absorbed by photosynthetic pigments, such as 480 nm, seems to have little effect on the systems control. Our analysis also confirmed the intuitive understanding that remaining respiration under low light could have low control on the CBB while alternate electron flow became influential under light saturation (S9 Fig). Using the model, the control of single components, such as photosystems, can also be investigated (S10 Fig).

## Model as a platform to test alternative mechanisms of state transition

We defined four functions representing the mechanism of state transitions with the PSII-quenching model [69] as the default. Therein, a higher fraction of PSII excitations is lost as heat in state 2. Three alternative models of state transition were tested: the Spillover Model, where state 2 induces PSII excitation energy transfer to PSI; the PBS-Mobile model, where PBS attach preferentially to PSII or PSI in states 1 and 2, respectively; and the PBS-detachment Model, where the increased $F'_m$ in state 1 is interpreted as the detachment of PBS [42, 69, 70]. We model the transition to state 2 under a reduced PQ pool while oxidized PQ promotes state 1. To test the general behavior of these mechanisms without limitation to a single parameter set, we systematically simulated each mechanism with a range of parameter values, typically varying each parameter within two orders of magnitude. Assuming an orange light intensity of 200 μmol(photons) $m^{-2}$ $s^{-1}$, we calculated the relative difference of PQ reduction under state 2 lighting to a model with no state transitions and the reduction of $F'_m$ from state 1 to state 2 (Fig 6B and 6C). In our implementation of the PBS detachment model, state 2 leads to the attachment of PBS to the photosystems. Because a small fraction of PBS is constantly detached in the default model, this reattachment may lead to a higher fraction of attached PBS and, therefore, a higher reduction of PQ. Overall, the PBS detachment model simulated a decreased PQ reduction in only 15% of simulations. The parameters in the PBS-mobile model modulate the attachment of the light-harvesting complexes to the photosystem. Therefore, depending on the mechanisms' parameters, either photosystem can preferentially receive excitations, resulting in the widest range of simulated PQ reduction states. The spillover and PSII-quenching models simulated consistent alleviation of the redox state, with the former having a lesser effect. All models could reproduce the expected decrease in $F'_m$ during state transitions (Fig 6D). However, the PBS detachment model showed the lowest mean response of all tested models. The PBS mobile model produced the largest variability in responses again.

We repeated the analysis of the PQ redox state for a range of light intensities (Fig 6D). The median alleviation was generally stronger at low intensities. Additionally, the PBS-mobile model produced more extreme increases in PQ reduction as well. For increasing light, all mechanisms showed lower effective alleviation. For example, the median alleviation of the PSII-quenching and spillover models decreased by 76 and 73% respectively when the light intensity was raised from 100 to 300 μmol(photons) $m^{-2}$ $s^{-1}$. For the strongest light, only the PBS-mobile model was able to alleviate redox stress. We also found that, under low light, carbon fixation decreased non-linearly when PQ reduction was lowered by the state transition models (Fig 6E). In high light, the carbon fixation rate stayed mostly constant in the spillover

model, even under redox alleviation, while the non-linear relationship remained in the PBS mobile model (S11 Fig).

## Model as a platform to test optimal light for biotechnological exploration

Cyanobacteria show potential as cell factories for the production of terpenoids from $CO_2$ or as whole-cell biocatalysts, which require different ratios of NADPH, ATP, and carbon. Several studies revealed that light availability is one of the main limitations of light-driven whole-cell redox biocatalysis [71]. With our model, we systematically analyzed the *Synechocystis* productivity for various light sources.

To identify potentially optimal light conditions and/or quantify the maximal production capacities for these exemplary processes, sink reactions were added to the model, and production was simulated with different light conditions (Fig 5B–5D). These sinks drained the required amounts of ATP, NADPH, and Fd necessary for fixing the required amount of $CO_2$ and producing one unit of the target compound. Additionally, it was necessary to add reactions that avoid overaccumulation of ATP and NADPH in case the sink was not sufficiently consuming both. The model simulates that NADPH production was highest under red (624 nm) illumination saturating around 800 µmol(photons) $m^{-2}$ $s^{-1}$. We also compared the simulated productions of isoprene and sucrose, which require different optimal rations of ATP and NADPH, 1.46 and 1.58, respectively. Isoprene production showed a stronger dependency on red-wavelength light, exceeding the production in blue light twofold, and did not saturate within the simulated light intensity range. Presumably, the involvement of Fd as a substrate further favors the usage of light preferentially exciting PSII. In recent work, Rodrigues *et al.* (2023) [29] measured isoprene production and used a genome-scale metabolic model to simulate cellular metabolism. Similar to our results, their predicted isoprene production rates followed the dynamics of the cell's growth rate (Fig 5E and 5F). Therefore, we agree with the assessment of Rodrigues *et al.* that isoprene production is not mainly governed by the differential excitation of photosystems but by downstream metabolism. On the other hand, simulation of the more ATP-intensive sucrose production was saturated at much lower light intensities and even decreased slightly under high light. These simulations indicate that the optimal light intensity could be lower for synthesis reactions requiring more ATP. It has also been suggested that the ATP:NADPH ratio is increased under blue light due to higher CET activity [72, 73]. However, our model did not show a benefit of more chlorophyll-absorbed light on the reactions involving ATP. Overall we found 624 nm light, to have the highest simulated production across the tested compounds and lights.

The light color and intensity showed pronounced effects in both cellular productivity and systems control. Therefore, we expected productivity under biotechnological overexpression to also be strongly light-dependent as it is affected by metabolic control. Our previous analysis showed that the control of PSII, FNR, and $Cb_6f$ on carbon fixation differed between light colors and light intensities (S12 Fig). Therefore, we simulated the steady-state carbon fixation rate of overexpression models where each protein's associated reaction was sped up by a factor of two (Fig 8). The overexpression of PSII led to an up to 40% increased carbon fixation rate under 440 nm light while showing little effect in orange light. $Cb_6f$ overexpression improved carbon fixation by 5% for orange light intensities under 100 µmol(photons) $m^{-2}$ $s^{-1}$ with diminishing returns for higher light intensities. Blue light illumination profited from the $Cb_6f$ up to the highest tested light intensity. The effect of FNR overexpression showed an almost inverted pattern of effects where carbon fixation was unaffected or even reduced under low light intensities. Only under high orange light intensities did FNR result in notably higher CBB rates. While these results were obtained for an unadapted cell, our model allows us to repeat such

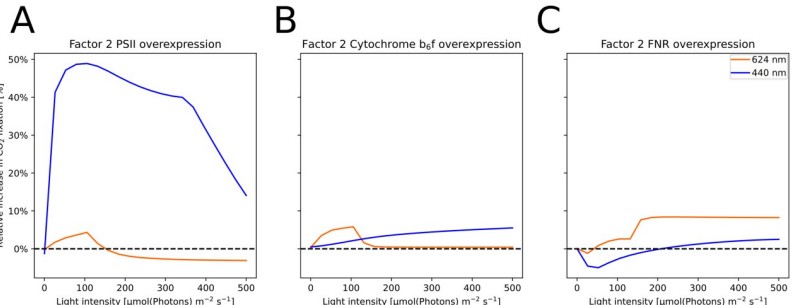

**Fig 8.** *In silico* **analysis of gene overexpression.** Simulated change in the rate of the CBB under twofold overexpression of PSII (A), FNR (B), and C$b_6f$ (C) depending on the irradiance. Simulations were performed under increasing intensities of either 440 nm or 624 nm Gaussian LED light. The change in CBB rate differs both between the light colors and the low and high light intensity regimes.

analyses with any adapted pigment composition (e.g. comparison of estimated $CO_2$ consumptions in S13 Fig).

## Discussion

In this work, we present the first wavelength-dependent mathematical Ordinary Differential Equation (ODE)-based photosynthesis model for cyanobacteria. The model contains all major processes involved in the *Synechocystis* photosynthetic electron flow, from light capture to $CO_2$ fixation [17] and a description of the respiratory chain embedded within the same membrane. Furthermore, cyanobacteria-specific mechanisms were implemented in the model, including state transitions and OCP-mediated NPQ [3, 39, 42, 74]. In contrast to other existing dynamic models of photosynthesis, our model takes pigment composition of the strain as an input and can simulate illumination within the full visible spectrum (400–700 nm). Hence, results obtained with our model provide insights into the intricate dynamics of the photosynthetic process under various light conditions.

Because the available data is too sparse to perform desirable post-regression analyses [75] we conducted a robustness analysis. We used Monte Carlo simulations to test whether the simulation results are sensitive to changes in our fitted parameters. Therein, we simulated 10,000 models in which we randomly varied parameters whose values were uncertain during parametrization. By varying parameters up to twofold, we did not find any set of parameters that better described our data for the selected multi-objective function.

The model was validated against published measurements of gas exchange rates (Fig 2F) and fitted to *in vivo* electron pathway fluxes and cellular fluorescence. The quantitative agreement with oxygen production rates supports our pigment-specific implementation of light absorption, which allows for a better assessment of the possible effect of photosystem imbalance [38, 62]. After parameterizing the model to reproduce the electron fluxes in the wild type, we used it to gain in-depth information on the system's behavior using *in-silico* mutants. Simulations of a Flv knockout mutant showed increased CET by NDH-1 under intermediate light (Fig 2C). It was reported previously that the proteins provide redundancy for alleviating redox stress [58, 76]. Furthermore, in the Flv mutant, flux from PSII is decreased due to lack of electron outflow to Flv (see S5C Fig). The decreased PSII flux is accompanied by raised NPQ under high light intensities.

Our calculated PAM fluorescence signal is composed of signals originating from both photosystems and PBS, with a similar contribution as in the previously published model [26]. We

employed this fluorescence estimate to fit a PAM-SP experiment inducing state transitions and OCP quenching (Fig 3A). We reached a qualitative agreement in the fluorescence dynamics, especially during the induction of OCP. Therefore, despite existence of more detailed models of OCP dynamics [31], we decided to keep our two-state implementation. The description of state transitions is challenging, as there currently is no literature consensus on the mechanism of state transitions [3, 42]. Therefore, we used our model to compare the implementations of four proposed mechanisms based on the cellular redox state and fluorescence.

The effect of the mechanisms strongly depended on the light intensity (Fig 6D). According to the literature, the transition to state 2 should rebalance the PQ redox state by oxidizing the pool [42]. Models of all mechanisms could produce this effect. However, the PBS detachment model failed to alleviate the PQ reduction in over 80% of simulations, especially under high light. Therefore, this mechanism seems unfit to describe the physiological effect of state transitions. On the other hand, while higher light intensities seem to reduce the effectiveness of the state transitions, the PBS mobile model, which is mechanistically similar to plant state transitions, retained the highest effect. In general, models with this mechanism consistently simulated the largest range of steady-state reduction states under parameter variation. Therefore, the targeted movement of PBS could provide the cell with high control over its electron transport. The significance of PBS movement has been debated, however [77–80], as has the spillover of energy between the photosystems [77–79]. It is noteworthy that considering solely the effect on PQ redox state, the implemented PSII-quenching model favored by Calzadilla *et al.* [42] does not have a significantly greater effect on the oxidation of PQ in our simulations. This limited PQ oxidation is in line with a model of plant photoinhibition where PSII quenching decreased PSII closure by ca. 10% [81]. Across all simulations, we found that the transition to state 2 was associated with a decrease in carbon fixation, particularly under low light (Figs 6E and S11). This tradeoff relationship was non-linear and differed slightly depending on the used state transition mechanism. Generally, if a state transition mechanism provided alleviation of redox stress, the PQ redox state improved by a higher factor than that by which carbon fixation decreased. Therefore, we find all mechanisms but PBS detachment physiologically beneficial, though the correct mechanism of state transitions and their impact on photosynthetic balance remains to be further evaluated.

We used MCA to systematically study the effect of light (intensity and color) and determined the systems control on carbon fixation considering varying illumination: solar illumination, a fluorescent lamp, and cold and warm white LEDs (Fig 4A–4D) of different intensities. The photosystems mainly controlled carbon fixation in simulations of low light intensity, which is in line with the limitation of light uptake and ATP and NADPH production as found in analyses of plant models [51]. Spectra with a high content of blue wavelength photons, which have been linked with an imbalanced excitation of PSI and PSII [37], showed a further increase in photosystems control. Indeed, blue light was found to increase PSII expression [72, 82], a cellular adaption possibly using this control. At higher light intensities, the maximum rate of carbon fixation became the main controlling factor. Thus, the strategy promising better productivity would involve increasing carbon fixation by e.g. additionally increasing the $CO_2$ concentration around RuBisCO [83], engineering RuBisCO itself [84] or introducing additional electron acceptors and carbon sinks such as sucrose, lactate, terpenoids or 2,3-butanediol [85–89].

With the implementation of the spectral resolution, our model could also simulate cellular behavior in high cell densities (e.g. bioreactors), where the light conditions might differ throughout the culture [90]. We show that lighting in the orange-red spectrum requires the lowest intensity to saturate the photosystems, with a warm-white LED showing the same efficiency as a fluorescent light bulb, an important consideration when calculating process costs (Fig 4E).

To showcase the biotechnological usability of this work, we analyzed the *Synechocystis* productivity for various light sources (Fig 5B–5D). Many experimental studies have investigated optimal light colors for the production of biomass or a target compound, with most studies agreeing that white or red light is optimal for cell growth but varying results for target compounds [29, 91, 92]. Especially the synthesis of light harvesting or protection pigments is regulated and strongly dependent on the light color [93–96]. These works point out that biotechnological production can be strongly improved using "correct" lighting. However, finding such optimal experimental conditions may be hindered by, for example, the active regulation of pigment synthesis-processes that could be overcome by cellular engineering. Using our model of a cell without long-term adaption, we may identify optimal conditions to aim for in cell engineering and experimentation. By simulating a target compound consuming the amount of ATP, NADPH and reduced Ferredoxin (Fd) necessary to synthesize the target compound from carbon fixation, we tried to estimate the maximum production potential without limitation by the CBB. We found that the simulated production of all three compounds was highest under red light illumination (624 nm). Sucrose production saturated at intermediate light and even showed slight inhibition under high light, while the simulated isoprene production, requiring reduced Fd and a lower amount of ATP, showed the highest requirement for light (no saturation at 1600 µmol(photons) $m^{-2}$ $s^{-1}$). Thus, the composition of energy equivalents seems to determine the optimal lighting conditions. NADPH production in particular seemed to follow a light saturation curve with maximum around 1600 µmol(photons) $m^{-2}$ $s^{-1}$. For the purpose of whole-cell biocatalysis, NADPH is often the only required cofactor for the reaction, while the generation of ATP and biomass are secondary. Studies have attempted to optimise NADPH regeneration through inhibition of the CBB, deletion of flavodiiron proteins, or introducing additional heterologous sinks for ATP, while at the same time trying to avoid oxidative stress [97–99]. Our simulations suggest that a switch in light color towards monochromatic red light may be a viable strategy to improve catalysis by matching the NADPH-focused demand of the sink reaction with an equally biased source reaction.

These results again support the need to test and optimize light conditions for each application on its own, as the stoichiometry of the desired process changes light requirements. Recently, two-phase processes have been used to increase titers in cyanobacterial biotechnology, arresting growth to direct all carbon towards a product [100]. Our model suggests that as a part of this process, changes in light color could be used to intentionally create imbalances in metabolism and direct flux to the desired product according to the energetic needs of the particular pathway.

Because our MCA showed that the systems control shifts with changes in the light, we inferred that biotechnological changes to the reaction kinetics might additionally change the light-dependency of cellular production. Therefore, we simulated the overexpression of PETC components with high, light-dependent control over the carbon fixation. Simulations at 440 nm blue light showed an up to 40% increased carbon fixation rate under the twofold overexpression of PSII. This result emphasizes the imbalance between photosystems that is known for blue illumination. Interestingly, while an increased $Cb_6f$ rate yielded an overall positive effect, overexpressing FNR below 200 µmol(photons) $m^{-2}$ $s^{-1}$ decreased carbon fixation by up to 5%. FNR is in competition with cyclic and alternate electron flow. Therefore, our simulation suggests that the electron pathways are more prone to become unbalanced under blue illumination. On the other hand, we found that carbon fixation under the highly effective 624 nm light profited most from increasing $Cb_6f$ under lower intensities ($<$100 µmol(photons) $m^{-2}$ $s^{-1}$) and FNR under high-intensity light, increasing carbon fixation by ca. 5% each. Accordingly, our model suggests the production in orange lighting can be improved by speeding up LET and improving the outflow of electrons into sink reactions.

To address the limitations of the current model, it is imperative to critically evaluate its underlying assumptions and identify key areas for improvement. For instance, with the current version of the model, we cannot predict the long-term cellular adaption governed by many photoreceptors [101, 102]. For each simulation, we assume fixed pigment composition and light absorption capacity, thus, analyzing a given cell state. Relevant cellular adaptions can, however, be used as new inputs according to experimental data. Also, rhodopsin photoreceptors can perform light-driven ion transport and, if found photosynthetically relevant, would be a useful addition to the model [103, 104]. Next, although our model considers the CBB as the main sink for energy equivalents, reactions downstream of the CBB, such as glycogen production [105], could pose additional significant sinks depending on the cell's metabolic state, necessitating further refinement of our model to accurately capture these dynamics. Additionally, further improvements of the currently significantly simplified CCM (Fig 2H) and photo-respiratory salvage functions could be beneficial, also due to the engineering efforts in building pyrenoid-based $CO_2$-concentrating mechanisms *in-planta* [106]. Photodamage may be a necessary addition to the model when considering high-light conditions, specifically PSII photoinhibition and the Mehler reaction [107] (see i.e. [30]). Finally, our model follows the dynamic change in the lumenal and cytoplasmic pH but is lacking the full description of *pmf*. An envisaged step of further development will be the integration of the membrane potential $\Delta\Psi$ into the model and simulation of ion movement, as presented in several mathematical models for plants [18, 108]. It would be moreover interesting to include the spatial component into the model, accounting for the dynamics of thylakoid membranes, as revealed by [109]. Thanks to our computational implementation of the model using the package `modelbase` [54], the model is highly modular, and the addition of new pathways or the integration of other published models (e.g. a recent CBB model [110]) should not constitute a technical challenge.

In conclusion, the development of our first-generation computational model for simulating photosynthetic dynamics represents a significant advancement in our comprehension of cyanobacteria-specific photosynthetic electron flow. While acknowledging its imperfections, our model has proven to be a versatile tool with a wide range of applications, spanning from fundamental research endeavors aimed at unraveling the complexities of photosynthesis to practical efforts focused on biotechnological optimization. Through a comprehensive presentation of our results, we have demonstrated the model's capacity to elucidate core principles underlying photosynthetic processes, test existing hypotheses, and offer valuable insights on the photosynthetic control under various light spectra. With further development and integration of experimental data, we hope to provide a reference kinetic model of cyanobacteria photosynthesis.

## Supporting information

**S1 Appendix. Detailed explanation of the model and further analysis.** Contains lists of all parameter values used in the model (Table A), all modeled reactions (Table B), initial conditions (Table C), parameters used to model light-adapted cells (Table D), and ranges of parameter variation during state transition model analysis (Table E). We also include explanations of the reaction kinetics and Gibbs energy calculations used, as well as further analyses of the model.
(PDF)

**S1 Fig. Schematic of the photosystem's modeled internal processes.** A: Photosystem II. The open reaction centers (RC) $B_0$ are excited by light (yellow bolt). The excited state $B_1$ can relax to $B_0$ by heat (*H*) and fluorescence (*F*) emission or perform photochemistry. The latter promotes the RC to the closed state $B_2$ and extracts one electron from water. Excitation of $B_2$ can only be quenched as *H* or *F*. Lastly, $B_2$ can reduce Plastoquinone (PQ) and enter the open state

$B_0$ again. Parentheses show the assumed state of the special pair chlorophyll $P_{680}$ (*D*) and electron acceptor plastoquinone A (*A*): excited (*) and reduced($^-$). B: Photosystem I. Light excites the open reaction centers $Y_0$. The excited $Y_1$ state can perform photochemistry by reducing Ferredoxin (Fd) and becoming oxidized to $Y_2$. We also consider a minor relaxation of $Y_1$ to $Y_0$ through *F*. The oxidized $Y_2$ is reduced by Plastocyanin (PC). Parentheses show the assumed state of the reaction center $P_{700}$.
(TIF)

**S2 Fig. Parameters of the model in Monte Carlo simulations.** We performed 10,000 simulations of the model with a subset of parameters being randomized: The 24 parameters marked as "manually fitted" in Table A in S1 Appendix were randomly varied within ±factor 2 (A) or ±10% (B). We drew independent randomization factors for each varied parameter in each model from a log-uniform distribution. We show the distribution of log-transformed parameter values used in the Monte Carlo simulations. 1.6% of simulations in B showed an improvement in all residual functions and a distribution of their parameters is overlaid in red. The red histogram is rescaled for visibility.
(TIF)

**S3 Fig. Steady-state fraction of open PSII reaction centers under different light intensities.** The model was simulated to steady-state under illumination with a fluorescence lamp spectrum at intensities between 0.1 and 700 μmol(photons) m$^{-2}$ s$^{-1}$. The open fraction was calculated as the fraction of PSII in non-reduced states $B_0$ and $B_1$ [50] (see S1 Fig).
(TIF)

**S4 Fig. Comparison of simulations performed with default parameters or a locally optimized parameter set.** A,B: Repetition of Fig 3A and 3B. C,D: Repetition of Fig 2F and 2G. Solid lines show simulations using the optimized parameter set. For comparison, the default simulations are shown with dashed lines. We optimized the parameter set by minimizing the mean of all residual functions including validation data. Additionally, we penalized the optimization score if any residuals worsened compared to the simulation with default parameters. The simulations with optimized parameters moderately improved the fit of the data but did not show new behavior or features. RMSE quantifies the residuals of the respective simulation with optimized parameters. The difference to the residuals of a model with default parameters is in parentheses.
(TIF)

**S5 Fig. Simulated fraction of reduced pools, lumenal and stromal pH and fluxes through ATP synthase, carbon fixation (CBB) and cyclic electron flow (NDH) for four *in silico* lines.** A-D: Per column, the models represent the wild type (WT) in saturating $CO_2$ (A) and ambient air $CO_2$ (400 ppm, B), a flavodiiron (*Flv*1/3) knockout mutant (C) and NAD(P)H Dehydrogenase-like complex 1 (NDH-1) knockout mutant (D). The levels and production fluxes of ATP and NADPH are shown as the primary output metabolites of photosynthesis, next to central redox carriers. The CBB flux in the last row is rescaled to the rightward axis for better visibility.
(TIF)

**S6 Fig. Calculated NPQ and PSII effective quantum yield (Y(II)) and the fraction of total excitations quenched as heat for four *in silico* lines.** A-D: The models represent the wild type (WT) in saturating $CO_2$ (A) and ambient air $CO_2$ (400 ppm, B), a flavodiiron (*Flv*1/3) knockout mutant (C) and NAD(P)H Dehydrogenase-like complex 1 (NDH-1) knockout mutant (D). The electron fluxes were calculated according to section S1.7 in S1 Appendix. The models

were simulated to steady state for light intensities between 0.1 μmol(photons) m$^{-2}$ s$^{-1}$ and 300 μmol(photons) m$^{-2}$ s$^{-1}$. Modeled conditions as in Fig 2A–2D.
(TIF)

**S7 Fig. Light pulses of different wavelengths differ in triggering fluorescence quenching.** The measurements were performed with Multi-Color PAM (Walz, Effeltrich, Germany). Low-intensity pulses (SP-Int = 1) affect the steady-state fluorescence (F) only weakly. With each pulse of 440 nm and 480 nm light, however, the F level decreases stepwise, pointing at fluorescence quenching, possibly through Orange Carotenoid Protein (OCP). The culture of *Synechocystis* sp. PCC 6803 was pre-cultivated in a conical flask on a shaker under cool white light (30 μmol(photons) m$^{-2}$ s$^{-1}$) at 23˚C to OD$_{750}$ = 0.2 (measured with Shimadzu UV-Vis 2600 spectrophotometer, Shimadzu, Kyoto, Japan). For the measurement, 1.5 mL culture was transferred to a quartz cuvette and dark-acclimated for 5 min prior to each measurement. During the measurement, a custom-made protocol was used with the following settings: Analysis mode: SP analysis; AL off; SP-int = 1 (500 s) / 20 (500 s); SP-color = 440 nm / 480 nm / 625 nm; ML-color = 625 nm.
(TIF)

**S8 Fig. Results of Metabolic Control Analysis (MCA) under illumination with near-monochromatic Gaussian LEDs.** We simulated the model under the lights in Fig 5A with a range of intensities from 80 to 800 μmol(photons) m$^{-2}$ s$^{-1}$ to steady-state. By varying the photosystem concentrations (A) and the maximal rate of the CBB (B) by ± 1%, we quantified their control on the CBB flux. Plots show the absolute control coefficients. The left graph shows the mean of both photosystems. Higher values signify stronger pathway control.
(TIF)

**S9 Fig. Results of Metabolic Control Analysis (MCA) performed for different light sources.** We simulated the model to steady-state with the lights in Fig 4A–4D at a range of intensities from 80 to 800 μmol(photons) m$^{-2}$ s$^{-1}$. By varying the protein concentration, maximal velocity, or rate constant of a reaction by ± 1%, we quantified their control on the CBB flux by calculating flux control coefficients. We display the absolutes of control coefficients as means within the following electron pathways: light-driven (A; PSI, PSII, Cytochrome $b_6f$ complex, NDH-1, FNR), alternate (B; Flv, Cytochrome *bd* quinol oxidase (Cyd), Cytochrome *c* oxidase), and respiration (C; lumped respiration, Succinate Dehydrogenase, NDH-2). Higher values signify stronger pathway control.
(TIF)

**S10 Fig. Control of photosystems and the CBB on metabolite concentrations of ATP, NADPH, and 2-phosphoglycolate.** Per column, we investigated the control of PSI (A), PSII (B), and CBB (C). We simulated the model under the lights in Fig 5A with a range of intensities from 80 to 800 μmol(photons) m$^{-2}$ s$^{-1}$ to steady-state. By varying the photosystem concentrations and the maximal rate of the CBB by ± 1%, we quantified their control on the metabolite concentrations. More positive/negative values signify a stronger positive/negative control of the respective PETC component. The photosystems control is generally highest under illumination within the chlorophyll absorption spectrum (405 nm, 440 nm and 674 nm), with PSII also having control in the red spectrum. The CBB has a generally high control until a critical light intensity is reached. Increasing PSI or CBB flux generally lowers ATP and NADPH concentrations and increased 2PG, PSII has the opposite effect. At high light, some control relationships become inverted.
(TIF)

**S11 Fig. Results of the systematic parameter variation of three state transition models.**
A-C: Simulations for the Spillover Model (A), PBS-detachment Model (B), and the PBS-Mobile model (C). Each point shows the result of a model run with varied parameters. The models were simulated for different light intensities (x-axis), and we calculated how the activation of the state transition mechanism affected the reduction of the PQ pool (y-axis) and the rate of the CBB (z-axis). For higher light intensities, the state transition has less effect on the PQ pool and CBB rate. Our implementations of the PBS detachment and mobile models can both lead to an additional reduction of the PQ pool. The rate of the CBB generally decreased with alleviation of the PQ redox state but also decreased under strong overreduction (see C). Only the PBS-mobile model simulates a decrease in PQ redox state under high light.
(TIF)

**S12 Fig. Results of Metabolic Control Analysis (MCA) for reactions with strongly light-dependent control.** We show the flux control coefficients of PSII (A), FNR (B), and Cytochrome $b_6f$ complex (C) under light variation. We simulated the model to steady-state using the lights in Fig 5A with a range of intensities from 80 to 800 μmol(photons) $m^{-2}$ $s^{-1}$. By varying the protein concentration, maximal velocity, or rate constant of a reaction by ± 1%, we quantified their control on the CBB flux by calculating flux control coefficients. Values above zero show a positive effect of increasing the reaction rate and values below zero show a negative effect. We selected these reactions for analysis Fig 8 because their control coefficients strongly varied between light intensities and colors.
(TIF)

**S13 Fig. Simulated steady-state rates of $CO_2$ fixation under monochromatic lights with varying intensites.** Simulations with default pigment composition (A) or with pigment compositions of *Synechocystis* sp. PCC 6803 grown under the respective light color (B). The adapted models were parameterized using pigment, photosystems, and PBS measurements. The models were then simulated to steady-state with the respective light condition, and the $CO_2$ consumption is shown. The $CO_2$ fixation rate is the lowest between 465 and 555 nm compared to all other tested conditions. Under 633 nm light, the highest $CO_2$ fixation rate is reached at the lowest intensity. Compared to the unadapted simulations, the efficient usage of red light and inefficient usage of blue light is more pronounced.
(TIF)

## Acknowledgments

We would like to thank Ilka Axmann and Marion Eisenhut for the initial conversations on the physiology of cyanobacteria that motivated the construction of this detailed model and David Fuente for the discussion on cyanobacterial absorption and pigment compositions. We also thank Marvin van Aalst for supporting code optimization and Dan Howe for helping with multiprocessing. We would like to extend our gratitude to Klaas J. Hellingwerf and Milou Schuurmans for sharing with us the original oxygen measurement data from their publication from 2014.

## Author Contributions

**Conceptualization:** Elena Kullmann, Anna Barbara Matuszyńska.

**Data curation:** Tomáš Zavřel, Jan Červený, Gábor Bernát.

**Formal analysis:** Tobias Pfennig, Elena Kullmann, Tomáš Zavřel, Andreas Nakielski, Jan Červený, Gábor Bernát, Anna Barbara Matuszyńska.

**Funding acquisition:** Oliver Ebenhöh, Jan Červený, Gábor Bernát, Anna Barbara Matuszyńska.

**Investigation:** Tobias Pfennig, Elena Kullmann, Tomáš Zavřel, Andreas Nakielski, Oliver Ebenhöh, Jan Červený, Gábor Bernát, Anna Barbara Matuszyńska.

**Methodology:** Tobias Pfennig, Elena Kullmann, Oliver Ebenhöh, Anna Barbara Matuszyńska.

**Project administration:** Anna Barbara Matuszyńska.

**Software:** Tobias Pfennig, Elena Kullmann, Anna Barbara Matuszyńska.

**Supervision:** Anna Barbara Matuszyńska.

**Validation:** Tobias Pfennig, Elena Kullmann, Tomáš Zavřel, Andreas Nakielski, Jan Červený, Gábor Bernát, Anna Barbara Matuszyńska.

**Visualization:** Tobias Pfennig, Elena Kullmann, Tomáš Zavřel, Anna Barbara Matuszyńska.

**Writing – original draft:** Tobias Pfennig, Elena Kullmann, Andreas Nakielski, Anna Barbara Matuszyńska.

**Writing – review & editing:** Tobias Pfennig, Anna Barbara Matuszyńska.

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
