## [Decision Letter · Decision Letter 0]

15 Dec 2023

Dear Dr Pfennig,

Thank you very much for submitting your manuscript "Shedding light on blue-green photosynthesis: A wavelength-dependent mathematical model of photosynthesis in Synechocystis sp. PCC 6803" for consideration at PLOS Computational Biology.

As with all papers reviewed by the journal, your manuscript was reviewed by members of the editorial board and by several independent reviewers. In light of the reviews (below this email), we would like to invite the resubmission of a significantly-revised version that takes into account the reviewers' comments.

I apologise for the delay on the decision but I was trying to secure a third review before the final decision. Both reviewers have raised major concerns with this study which would have to be addressed for acceptance of this article into PloS Computational Biology. All comments from both reviewers, especially reviewer 2, need to be addressed if this paper is resubmitted.

Best regards

David Lea-Smith PhD

We cannot make any decision about publication until we have seen the revised manuscript and your response to the reviewers' comments. Your revised manuscript is also likely to be sent to reviewers for further evaluation.

Sincerely,

David Lea-Smith

Guest Editor

PLOS Computational Biology

Jason Haugh

Section Editor

PLOS Computational Biology

Dear Dr Pfennig

I apologise for the delay on the decision but I was trying to secure a third review before the final decision. Both reviewers have raised major concerns with this study which would have to be addressed for acceptance of this article into PloS Computational Biology. All comments from both reviewers, especially reviewer 2, need to be addressed if this paper is resubmitted.

Best regards

David Lea-Smith PhD

Reviewer's Responses to Questions

**Comments to the Authors:**

Reviewer #1: The model described and used in this manuscript is not perfect and it suffers from being too simple (my opinion). However, it is the first kinetic ODE-based model for cyanobacterial photosynthesis, as far I know. Thus, it should be published, but the manuscript needs to be improved to be accepted for publication, especially in description of the model.

My detailed comments:

Introduction – The literature is not up-to-date. Following works should be cited:

Stirbet et al. (2019; Chlorophyll a fluorescence in cyanobacteria: relation to photosynthesis. In: Mishra A.K., Tiwari D.N., Rai A.N. (ed.): Cyanobacteria – From Basic Science to Applications. Pp. 79-130. Academic Press, London) – in places where cyanos and their ChlF is mentioned

Stirbet et al. (2020; Photosynthesis: basics, history and modelling, Annals of Botany 126, 511-537) – in places where modelling of photosynthesis and of ChlF is mentioned

Lazar et al. (2022; Light quality, oxygenic photosynthesis and more, Photosynthetica 60(SI), 23-56) – in places where light spectrum utilized by cyanos is mentioned

L150 – LET is not a circuit. A better wording should be used

Legend to Figure 1 – It is strange to consider Cyd, COX, and Flv as components of WWC. In your logic, also PSII and PSI should be parts of WWC. In fact, reactions starting by formation of superoxide from reduction of oxygen (the Mehler reaction, see review of Asada 2010, PCP) are the initial parts of the WWC. Reactions of oxygen reduction, accompanied by usage of protons, to water, as occurring in Cyd, COX, and Flv are not parts of the WWC, as far I know. The authors must check it.

There is no FQ in the figure, however, it is mentioned in the legend.

NDH-1 is described as NADPH dehydrogenase 1, however, it accepts electron from ferredoxin in the figure, thus, it cannot be NADPH-dehydrogenase but it is NADPH-dehydrogenase-like (NDH-like) complex, see related literature.

Figure 1 – I guess that in the case when Flv is not functional and flux from PSII is decreased due to lack of electron outflow to Flv, the decreased PSII flux is accompanied by increased NPQ or similar process. Is it right? If yes, it should be mentioned and shown.

L153 – Change “RuBisCO oxygenation” to “RuBisCO oxygenation function”

L162 – Even if it is not important, you should mention why you set the maximal differences differently. What is the reason for this?

L168-171 – If by directionality you mean that the reaction is only forward or reversible reaction (as implied from the sentences) than, the reaction can be only forward or reversible no matter if the Gibbs free energy difference is positive or negative. Thus, the reasoning of this sentences seems to be wrong and further, nowhere in the text or in the supplement any calculation of the Gibbs free energy difference is shown. Thus, you should rephrase it. And if you uses this approach, you should mention that it was used in your previous papers, not in the modelling papers generally.

L193-194 – How can reduced PQ pool molecules quench ChlF? As far I know, only oxidized PQ molecules were reported to quench ChlF. You probably mean that reduced PQ pool causes a switch of PSII to a state of PSII, which quenches energy by heat dissipation. It should be clarified and/or more described.

Equation 2 – Exact definition of FPSI, FPSII, and FPBS must be provided. What model variables (states of the model) were considered to emit florescence and with what quantum yields or what are values of related rate constants (for fluorescence emission)?

General comment to the model description in the supplement: the model is not described adequately well. For example, why there are symbols “-.” or “!-.” in table S3 instead of standard arrows. Further, the equations should be written by standard symbols and the reactants by their standard names, not by the symbols used in the model code – in this way you are kidding with the readers, since not all readers are familiar with such type of description. You must simply pay more time to a proper model description so that it can be unambiguously understood and reproduced even by a beginner in the field.

L222 and other places – The same curves as named by you as “PAM curves” can be measured by a non-PAM fluorometer. People usually mismatch the denotations and meaning. PAM fluorescence is method for detection of any ChlF signal (with measuring flashes or/and with actinic light or/and with saturation pulse). But what is unique in your experimental data, is application of the saturation pulses, which enables to calculate related quantum yields, which you did not calculated, unfortunately. Thus, the correct name of the method is the saturation pulse method (see some review papers/chapters by Uli Schreiber, inventor of PAM).

L225 – Change “Our simulated steady-state …” to “Changes in our simulated steady-state …”.

L229 – What is Bo and B1 in the legend of Fig. S2? It is related to my general comment on a poor model description.

L242 – Yes, ”LET stagnates”, but it would be better to write “CET stagnates”

L255 – What “measuring protocol discrepancies” yare writing about? Authors of the referred paper are authors of this manuscript, so there should be no discrepancies.

L260 – Why do you use a new term “Dark Peak Fluorescence”, when the Fm term defines exactly what it is?

L261 – What is QSS? It must be defined.

L263-264 – When you know the reason, have you checked the opposite scenario? Why not?

L274 – What do you mean by “We slightly perturbed …” How much is slightly? What was maximal relative change of the model parameter (rate constant) to calculate the control coefficients? The are some rules in MCA to do it and you should mention it.

L277 – “usually”? I would say “always”.

L283 – Is reference to Fig. 3c correct? Should not be here also reference to Fig. 3b?

L294 – Change “more reduced PQ pool” to “more reduced PQ pool in state 2”

L296-297 – This description of PBS-mobile model is wrong. When there is set state 1 (state 2), PBS is attached to PSII (PSI).

And

L297 – There should be “PBS detach from the photosystems II” instead of “PBS detach from the photosystems”.

Further, according to your statements and schemes in Fig. 4A, you assumed attachment of PBS to PSI in states 2 in the PSII quenching model, spillover model, and in PBD-detachment model, which was not assumed in the referenced review (30).

Since these are significant, mistakes/differences from original referenced suggestions, the authors must check, if these mistakes are only in the text and the simulations were done correctly or not. If simulations are wrong, they must be corrected, including related conclusions.

L300 – Why do you write in “Dark Steady-State Fluorescence” high initial letters?

L302 – Why do you write in “Peak Fluorescence” high initial letters?

Sections started at L269 and L313 explore effects of different excitation wavelengths on model outputs but in fact, nowhere is mentioned, how (mathematic formulas) different spectral excitations affect the system function. This must be clearly described in the text or in the supplement. In connection to that, for example, the legend to Table S2 writes about “light attenuation” but it is nowhere described how it is treated mathematically, which, however, must be done.

Table S1 – Is the “standard electrode potential” really the correct term? What about midpoint redox potential? And further, the redox potentials are always determined for redox couple, for example, for Qa/Qa- couple the midpoint redox potential is -140 mV. Thus, it must be corrected. Further, in section “kinetic constants” the constants are termed as rates, which is wrong, since they are rate constants not rates. Rate has the same meaning is flux through a reaction.

L374-375 – A role of Ribisco is tressed on the basis of caclulation of the control coefficient of Rubico, which might be misleading, since sedoheptolose bisphosphatase has been found to have the highest control for CO2 assimilation rate (papers by Poolman in 2000 and 2001).

A last question. How does the model consider the high PSI/PSII ratio known for cyanos when PSI as such is not considered in the model?

Reviewer #2: Comments on "Shedding light on blue-green photosynthesis: A wavelength-dependent mathematical model of photosynthesis in Synechocystis sp. PCC 6803" by Tobias Pfennig et al. submitted to PLoS Computational Biology.

The authors present an ODE-based model of photosynthesis of the cyanobacterium Synechocystis sp. PCC 6803, primarily focussed on the electron transport chain but also including (simplified) adjacent reactions (ATP usage, carbon fixation).  As a (nonlinear) ODE-based model the model requires knowledge of a substantial amount of kinetic parameters. The model follows traditional methods of kinetic modelling, in particular parameters are primarily based on literature values.

Models such as the one presented here are suitable to advance understanding of the cyanobacterial electron transport chain, including its dynamic properties. The latter puts kinetic models apart from stoichiometric constraint-based models that typically require far fewer parameters, but are also far more limited in their scope.

While I recognize the importance models such as the one presented here, I think the present model (and its description) can be substantially improved. In particular, I think the two main shortcomings of the manuscript are (1) the results should clearly demonstrate novelty and biological significance. Currently, the main achievements are that the model conforms reasonably to some available data (which in itself is not a distinguished result) and the results beyond that are inconclusive (state transitions) or very speculative (biotechnology). Secondly (2), model construction, the underlying assumptions, and in particular parameter estimation are poorly described.  

Estimating kinetic parameters is a major challenge for kinetic models, and the authors will surely be familiar with Von Neumann's elephant: "With four parameters I can fit an elephant, and with five I can make him wiggle his trunk". Here, the trunk certainly wiggles, but we do not know how the parameters were determined. 

More specifically: the text makes several strong claims for the achievements of the model "Our model accurately predicts the partitioning of electrons through four main pathways, O2 evolution, and the rate of carbon fixation. Additionally, it successfully captures chlorophyll fluorescence..." (in abstract). As far as I can see, these strong claims are not substantiated by data. 

Moreover, some of the claims are contradictory (and likely incorrect, see below for further examples). For example, in the caption of Figure 1: "Our model correctly predicts the proportion oflinear electron flow (blue) to four other pathways to be around 60% [39] in wild type (WT)". Later, it is stated that (line 231): "We parameterised the flux through the CET to yield approximately 35 % of the total PSI electron flux [39]."

Was the model parameterized according to the data from ref [39] or is it a prediction? 

According to which criteria/data was the statement  "Our model accurately predicts the partitioning of electrons through four main pathways, ..." evaluated? 

In line 253 we are informed that "... parameters controlling the fluorescence composition, state transitions, NPQ, and CET were *updated* to get the best agreement between our simulated trace and the original data 255 (Fig. 2a)". (emphasis by me). 

What does "updated" mean exactly? According to which methods (adjusting values manually, using a fitting algorithm)? 

Is the model overfitted? (i.e. many potential updated parameters would give an equally good fit)?

If yes, how were the final parameters chosen? 

These examples are not exhaustive, but rather illustrate the kind of problems that exist with respect to the handling of parameters. I consider such issues to be of high importance when evaluating a kinetic model that claims predictive value.  

It is further noted that the remaining section (Fig 3 onwards) are largely computational/speculative and not based on any further measurements or available data. 

My overall suggestion is that the "Methods" must have a dedicated section that describes the parameterization of the model. Vague statements, such as "We derived kinetic parameters from literature measurements, e.g. in vitro kinetics, fluorescence, or gas exchange rates" (line 139) are *not* sufficient. 

In general, model description should be expanded. It is a fairly straightforward model (this, in itself, is not a bad thing), almost all processes involved have been described in far more detail (in particular core pETC, here PSII and PSI are described by one mass-action equation, respectively).

Also some descriptions sound more elaborate that the model actually is  (L154, "we also considered a simple Carbon Concentrating Mechanism (CCM) implementation by ...", not described incorrectly in the main text, but the "simple CCM" is really just multiplying the concentration by a factor 1000. Why 1000 exactly? The cited Ref is far less specific).

The Method section would generally benefit from a clearer description of the model, e.g. how many reactions, how many parameters, etc ...? Even when looking at the Supplemental Text it is difficult to extract this information (not helped by miniscule font in Table S1 and terms like "visually fitted", whatever that may mean). 

Furthermore: whenever there are statements that claim "the model accurately predicts", or "model can accurately capture" (Cap. Fig S1) or "The model was validated against published measurements ...", it must be made clear how the correctness of the prediction was established, whether the data has been used in model parameterization. And if yes, to what extent the model is overfitted. (and what does "accurately capture" mean? The curve is either fitted, i.e., parameters were adjusted to match the curves, or it is a prediction, and the parameters were sourced elsewhere. I do not know what "capture" refers to in this context).

SPECIFIC COMMENTS BY LINE

L16: "iit is crucial to investigate their intricate interplay with the environment in a particular light". (intensity? quality?)

L19: Spelling: "an ordinary differential equation-based model for cyanobacterium Synechocystis sp. PCC 6803"

L20: "Our model accurately predicts the partitioning ...". Substantiate claim!

L21: "Additionally, it successfully captures chlorophyll fluorescence signals ...": what does it mean that a model "successfully captures" something? How is "success" defined in this context? 

L29: Keywords: Avoid keywords that are already in the title

L49 and following: A lot of space is devoted to explaining that prokaryotes lack organelles that are found in plant cells. On the other hand, methods like PAM are used without almost any explanation. It is clear that (most) of the authors come from the plant field, but the introduction should be aimed at a broad readership. (The section here reads a bit like an undergraduate term paper: "cyanobacteria can perform photosynthesis and respiration, allowing for auto-, hetero-, and mixotrophic lifestyles [15]. But, due to the lack of mitochondria, cyanobacteria ...". Also everything is very plant-centric).

L136: "A comprehensive list of all the model parameters utilised in this study, ..., is provided in Table S1 and Table S2". This section is insufficient. Model description should be in the main text. How many variables, how many parameters, etc ...

L143: Providing a github is very good, but the main text and Supplement should also be sufficient to fully understand the model. For example, a list of reaction rates in the Supplement would be helpful (currently Table S3 in the pdf seems to be garbled and is very difficult to read)

L165: Why was mass action kinetic linked to (known) regulatory properties? Whether MA is a good/poor approximation is not only determined by whether the reaction is regulated. How were rate equations motivated in general, given that many elements in the pETC have been modelled in great (biophysical) detail. Statements like "because parsimony" are too vague.  

L171, Eq (1): what happened to the stoichiometric coefficients here? 

L175: I do not understand how that the equilibrium equality can be violated by anything described here. Please explain.

L178 "This implementation enabled us to simulate various light-adapted cells by updating the parameters corresponding to measured pigment composition and ...". The description is too vague. Also the Supplement is not very helpful. Implementing different spectra is one of the main "selling points" of the model, the description should be far more specific. 

L219ff, Eq (3): I am not convinced that summing of absolute values is a good idea (it is not commonly used, for good reasons). Processes with more elements will get higher values (you primarily count elements/subprocesses). Why did you not use the median or mean? 

L225: "Our simulated steady-state O2 rates are in qualitative agreement with the experimental data (Fig. S1).". I do not find Fig. S1 very convincing. Do either the data or the simulations have error bars (e.g. the latter due to parameter uncertainty). I see a light dependent increase in O2 evolution, as expected, but almost any model would predict that, and the simulations are not very close to the data. 

L323: "We parameterised the flux through the CET to yield approximately 35 % of the total PSI electron flux [39].". How was this done, which parameters? ("Flux" is not a parameter).  Fig 2 Caption: "The simulations were adapted to the given strain by changing the parameters corresponding to pigment composition". How were simulations "adapted"? Were parameters changed? Which

parameters? By which method/criteria/algorithm?

L253: "and parameters controlling the fluorescence composition, state transitions,254 NPQ, and CET were updated to get the best agreement between our simulated trace and the original data255 (Fig. 2a).". There needs to be significantly more detail what terms like "updated" actually mean, and which information was used to parameterize the model and what is a "prediction".

L258 "Our simulation predicted especially well the transition to state 1 in blue light and the NPQ action and relaxation". Was that after the parameters were "updated"? 

Fig3: Labeling is a bit of a mess. There are capital letters for subplots, but also lower case letters. More importantly, Fig 3c (?) shows CO2 consumption as a function of light intensity. Net CO2 consumption should correspond to growth rate (in the absence of excretion of products), but the curve does not look like any growth rate ever reported. I note that the rate is measured relative to g(Chl), but g(Chl) is assumed to be constant in the simulations. How do the authors reconcile the statement that the model "correctly (!) predicts the rate of carbon fixation" with the fact that no measured rate of carbon fixation I am aware of looks even qualitatively like this curve. 

L331ff: The entire section is very speculative, with little grounding in data. 

L335: "In this work, we present our mathematical ODE-based photosynthesis model as a quantitative in silico tool with high predictive power and customisability." The "high predictive power" has to be substantiated. Currently it is not. 

L340 "our model can simulate illumination within the full visible spectrum (400 nm to 700 nm), also accounting for the pigment composition of the strain". But pigment composition and g(Chl) per cell will change under different spectra and intensities. Can the model predict that? If not, how can the model make predictions about, e.g. optimal conditions for biotechnology? If the pigment composition has to be measured before any simulation, the predictive values is limited. 

L347: "The model was validated against published measurements of gas exchange rates and electron pathway348 fluxes (Fig. S1). The good estimation of oxygen production rates supports ..." Do we look at the same Fig S1? I do not find the figure very convincing. How was the "validation" performed exactly?

L351: "Our simulations show the dominance of LET (ca. 60 %; [39]), ..." Compare with: "We parameterised the flux through the CET to yield approximately 35 % of the total PSI electron flux [39]." The constant obfuscation about what is a prediction and what was fitted is a MAJOR problem of this manuscript.

L361: "Our simulations support a PSII quenching model [62] rather than PBS detachment and spill-over, as suggested previously [30]". The simulations are inconclusive, but the PSII quenching is also widely accepted. And [30] is broad discussion and does not argue that the PSII quenching model is wrong. 

L400: "Our model considers the CBB as the main sink for energy equivalents which might not be the case for cells with higher growth rates, where the production of other biomass components can become more significant (e.g. glycogen [84])." How do you assume "production of other biomass component" happens without the CBB (or with a reduced CBB)? 

L433: Several authors are not listed in the "Authors contributions". 

Overall summary: Any claims of predictive power have to be substantiated. The part on biotechnology is too preliminary to be considered a "result".

**Have the authors made all data and (if applicable) computational code underlying the findings in their manuscript fully available?**

Reviewer #1: Yes

Reviewer #2: **No: **Code is available, but parameterization is unclear.

PLOS authors have the option to publish the peer review history of their article (what does this mean?). If published, this will include your full peer review and any attached files.

Reviewer #1: No

Reviewer #2: No
---

## [Decision Letter · Decision Letter 1]

18 Apr 2024

Dear Dr Pfennig,

Thank you very much for submitting your manuscript "Shedding light on blue-green photosynthesis: A wavelength-dependent mathematical model of photosynthesis in *Synechocystis* sp. PCC 6803" for consideration at PLOS Computational Biology.

As with all papers reviewed by the journal, your manuscript was reviewed by members of the editorial board and by several independent reviewers. In light of the reviews (below this email), we would like to invite the resubmission of a significantly-revised version that takes into account the comments from reviewer 2. It is critical these concerns are addressed before publication is possible.

We cannot make any decision about publication until we have seen the revised manuscript and your response to the reviewers' comments. Your revised manuscript is also likely to be sent to reviewers for further evaluation.

Sincerely,

David Lea-Smith

Guest Editor

PLOS Computational Biology

Jason Haugh

Section Editor

PLOS Computational Biology

Reviewer's Responses to Questions

**Comments to the Authors:**

Reviewer #1: The authors answered all my questions/comments and revised the manuscript accordingly.

Reviewer #2: Review: Shedding light on blue-green photosynthesis: A wavelength-dependent mathematical model of photosynthesis in *Synechocystis* sp. PCC 6803 by Pfennig et al. 

The authors have made extensive changes to manuscript, as required. The manuscript has clearly improved. The current version would have been a suitable first submission (as the actual first submission was defective on several levels). 

I have evaluated the manuscript carefully and have a number of comments, also taking into account the response letter of the authors. 

Comment #1: I am glad to see that claims about predictive power have been removed. "Upon reviewing it, we see that you're absolutely right. The word “predict” should not be used" and   "... we incorporated the recurring feedback that we are making too strong claims and have used more humble descriptions". 

I want to emphasize that my criticism was not about a lack of "humbleness" in the descriptions, but about (a lack of) scientific truthfulness/correctness.

One example: I noted the inclusion of a (very primitive) carbon concentration mechanism by multiplying the value of CO2 by 1000. The value was cited as derived from the literature. Now I learn that you actually tested different values and  "we could see that an increase of the intracellular partial pressure by factor 1000 could reproduce the dynamics of O2 evolution ...".

I do not recall that this was noted anywhere in the original manuscript. Instead you claimed to quantitatively predict O2 evolution. Such claims are not acceptable. I note that the issue is (almost) resolved in the current version, but I still hope that the senior authors discuss with the junior researchers how to present scientific results and how to properly back up claims about "predictive power". 

*This is no minor issue*. Readers must be able to trust researchers not to make misleading claims about predictive power of models, in particular in interdisciplinary work (the first submission failed in this respect).

Comment #2:   The authors write: "We disagree with Reviewer 2's statement that it is not a distinguished result itself to create a mathematical model that conforms reasonably to some available data nor ...." 

My sentence was actually part of a wider quote (that I deleted to shorten the review), taken from James Bailey [Mathematical Modeling and Analysis in Biochemical Engineering.... Biotechnol Progress, 14: 8-20 (1998). https://doi.org/10.1021/bp9701269]: "There is a zoo of mathematical models in the biochemical engineering and mathematical biology literature. Many of these appear, particularly to the naive reader (and sometimes to the sophisticated one), to have little purpose other than calculating numbers which conform reasonably to experimental data. This is, in itself, not a distinguished endeavor; it is not particularly difficult, and it teaches little."

The quote closely ties to von Neumann: "With four parameters I can fit an elephant, and with five I can make him wiggle his trunk".

Your model is indeed an example of this phenomenon: the model goes through a series of ad hoc adjustments that make it conform to data and suit your particular data. We already saw the factor 1000 above (fitted to give the best possible result). 

Further: "As literature light response curves suggested that we overestimated the photosynthetic yield, we have introduced a light conversion factor that describes the generation of excitations from 50% of absorbed photons" ... which is yet another ad hoc factor.

Further: "Because our model previously simulated a sudden decrease of cyclic electron flow under high light which was not consistent with literature growth curves we changed the description of NDH-1 mediated cyclic electron flow".

And so on ...

While you can do all that, you should also be aware that such adjustments to fit the present data have severe implications for the general validity and for the the predictive power of any model. What makes you certain that the "light conversion factor" remains the same across conditions? And if it is not constant, what does this mean for the model to be used to explore different conditions (as explicitely claimed as a goal in the manuscript), short of re-parameterizing for each condition (such a requirement would again severely limit the utility of the model)?

I note that you also already envision further adjustments if needed, i.e., from the response letter: "Therefore we decided to describe reactions using MA kinetics (modified for simplification) unless a different description was necessary to reproduce desired behaviour." 

Comment #3: The quote above continues to highlight roles of models beyond fitting, as you also point out ("Secondly, prediction and forecasting are only one of many goals of computational modeling, next to enabling understanding of complex systems and hypothesis testing .."). 

I fully agree with that. Most models (and some of the best ones) are not about predictions but are, for example, about explaining/understanding phenomena. However, it was your choice in the initial submission to claim predictive power.

I would encourage the authors to think carefully about what can be learned from the model in its current stage. That is there a profound new biological insights the model brings, other than being a "platform" or that models are useful in general?

Is there any genuine biological insight that can be drawn from the model in its present state? What do I learn about the biology of cyanobacteria that I was not able to learn without the model?

This is a serious question and all modellers should ask this question during writing. In the present case, I note that many of the potential genuine (and specific) results are rather speculative or inconclusive (state transitions, biotechnology).

Comment #4: I do have a major issue with the claim that "In the context of biochemical kinetic modeling such as this one, the justification for manual parameterization over sophisticated algorithms lies in the nuanced understanding and expert knowledge of the system under investigation."

I am fairly familiar with parameter estimation. Algorithms for parameters estimation do indeed allow for prior information ("nuanced understanding and expert knowledge"), they provide a clear understanding about the errors involved. Importantly they also provide information about "identifiability", i.e., which set of parameters are actually constrained by the data .

If you claim (as currently in the main text, line 161) that "To avoid overfitting the parameters to a particular experimental set-up, we avoided using sophisticated fitting algorithms and instead ...",  and "While fitting algorithms may offer automation and potentially expedite the parameterization process, they often lack the interpretability and domain-specific insight that manual parameterization provides", then please provide appropriate benchmark studies (as a reference) that show shortcomings of such "sophisticated fitting algorithms" over manual fits, and that manual fits avoid overfitting (as opposed to algorithms). 

I do not know of any such study (and know of plenty that demonstrate the contrary). 

Also: "Additionally, manual parameterization offers flexibility in addressing model complexities and adapting to unique experimental conditions or system variations that may not be adequately captured by automated algorithms."

Which specific "model complexity" and "unique experimental conditions" are you referring to?

Generally:  The introduction of ad hoc parameters, such as "we have introduced a light conversion factor" and others, are *textbook examples* of overfitting to suit a particular dataset. 

Comment #5: In my opinion any claim about *quantitative* agreement necessarily requires the use of appropriate measures to assess agreement. None of your analysis contains any such analysis of errors (neither with respect to the model, nor with respect to the data). Yet you frequently claim that the model is quantitative. 

There are rather oxymoronic sentences close to each other, such as "By harnessing the power of mathematical modelling, we seek to provide a quantitative framework to test further hypothesis ..." and "We do not provide quantitative measures to assess the quality of fits as PAM curves were fitted manually". 

This again also applies to figure S3: "Using the same fitted parameters, we can also reproduce the quantitative behavior of cells grown under 633 nm monochromatic light (S3 Figure) .. "

Qualitative yes, quantitative: no. If you want to claim quantitative agreement, you must put in the required quantitative analysis.

(and I still don't find Fig S3 very convincing, even from a qualitative perspective, let alone quantitative). 

Comment #6: Overall figure quality is poor. I don't know if this is because of the downloaded pdf (figures have rather poor resolution in the pdf). Also fonts are too small and difficult to read (see e.g. Fig 5B). 

In general, there are only few figures in the main text. The text often refers to Suppl. Figures for key results, e.g. S3. Some of these results should be shown in the main text, if they provide relevant information. 

Comment #6: Further on figures. Line 321ff: "Changes in our simulated steady-state O2 evolution rates are in quantitative agreement with the experimental data, during low light and exceed measured rates under light saturation by ca. 20 % (Fig 2C)".

Where is any "quantitative agreement" shown? How does Figure 2C relate to this statement. 

Same for: "The model was validated against published measurements of gas exchange rates (Fig 2C) and ...."

Comment #7: I don't know what Figure S3 shows.

The caption says: "S2 Figure. O2 production under light intensity variation in vivo and in vitro."The axis says something else. What is shown there? CO2 or O2? Something else?

Also: It is noted that the simulations and the measured values do not even *qualitatively* agree. The shapes are entirely different. You then define a threshold (ambient CO2) and claim "the intracellular partial pressure by factor 1000 could reproduce the dynamics of O2 evolution for ambient CO2 concentrations and above".

This collides with your claim to "we seek to provide a quantitative framework to test further hypothesis" and is exactly the kind of ad hoc choice described above. Such descriptions make me very sceptical about the general quantitative power of the model. You pick and choose to suit your narrative. 

What would stop you to claim "validation" if the points had matched over a wider interval? And If a different dashed line would have been necessary, why not choose a different threshold above which the model fits the data?

If one aims to answer a specific question or hypothesis, such choices may (sometimes) be acceptable, if clearly discussed and justified. But I do not think such an approach can lead to a general quantitative framework, as claimed here ("a comprehensive framework, as a validation of the accuracy of ourcurrent understanding" ... "developing a generic model with validity for more than one condition").

Comment #8: I assume equation S49 is incorrect, right?

Comment #9: The model has 27 reactions, the table only lists 22. What are the remaining reactions?

Further comments (in no particular order): 

- Line 453: "In line with the recent work by Rodrigues et al. (2023) [29], our simulated isoprene also follows the measured cellular growth rate as predicted by their stoichiometric model analysis (S13 Figure)."It follows the growth rate, but not the actual production rate (which is not surprising, since the equation for production is similar to the rate equation for growth).

- We have at least one russian reaction: "the reaction doesn’t involve free elections, ..."  (line 1351)

- "However, we disagree with the claim that the model does not capture (chlorophyll fluorescence) dynamics properly. We have captured the trend in both steady-state fluorescence and Fm levels, ..." 

I never claimed that the model does not  "capture properly". I mostly noted that I do not know what "capture properly" means, given the lack of any quantitative analysis of any deviations between data and model.

- "Because of this, we decided to remain with the manual fitting with a focus on qualitative reproduction of the data."Your text actually claims the opposite. You claim a general quantitative framework.

**Have the authors made all data and (if applicable) computational code underlying the findings in their manuscript fully available?**

Reviewer #1: Yes

Reviewer #2: Yes

PLOS authors have the option to publish the peer review history of their article (what does this mean?). If published, this will include your full peer review and any attached files.

Reviewer #1: **Yes: **Dusan Lazar

Reviewer #2: No
---

## [Decision Letter · Decision Letter 2]

6 Aug 2024

Dear Mr. Pfennig,

Thank you very much for submitting your manuscript "Shedding light on blue-green photosynthesis: A wavelength-dependent mathematical model of photosynthesis in *Synechocystis* sp. PCC 6803" for consideration at PLOS Computational Biology. As with all papers reviewed by the journal, your manuscript was reviewed by members of the editorial board and by several independent reviewers. The reviewers appreciated the attention to an important topic. Based on the reviews, we are likely to accept this manuscript for publication, providing that you modify the manuscript according to the review recommendations.

Can you please address the minor comments from the reviewer and once this are completed I will recommend the paper for acceptance.

Sincerely,

David Lea-Smith

Guest Editor

PLOS Computational Biology

Jason Haugh

Section Editor

PLOS Computational Biology

Can you please address the minor comments from the reviewer and once this are completed I will recommend the paper for acceptance.

Reviewer's Responses to Questions

**Comments to the Authors:**

Reviewer #2: I only have a few further comments.

1. I am glad to see that the claim that manual fitting is superior to a rigorous computational approach is no longer maintained. This claim did not hold much water. 

2. I would still prefer to see an analysis of error (error bars) in Fig. 2F as quantitative agreement is claimed. Error bars are a core of the scientific method and are not an unnecessary inconvenience (at least when quantitative agreement is claimed). 

2b. I am actually slightly suspicious about Fig 2F. In a model that is not over-parameterized it is rare that a curve lines exactly on the measured points. I suspect that because you fit electron flow (Fig 2A) and electron flow is stoichiometrically related to O2 evolution (LET minus AEF) you implicitly also fit the O2 evolution (with a deviation at higher light intensities). I do not recall the original publication, but often also experimental works calculate the values this way (that is, the O2 and the electron flow might not have been independently measured in the original publication. Might be worth checking).  

3. The MC sampling is not a replacement for parameter fitting. In particular, when using large intervals (e.g. factor 2 or more) you essentially run the model with random parameters. It is indeed unlikely that those improve upon the original parameterization (as you indeed found out). Small deviations are far more likely to find (local) improvements (that is why step size is important in, for example,  evolutionary algorithms, large steps just randomize everything). 

4. Line 171. You check if the parameterization is close to the Pareto front. But how do you know the Pareto front if you do not have the "best" parameterization (also a brief description in the Materials and Methods would help, not everyone will be familiar with what you actually ("checking if the model is on the Pareto front"), and I am also more guessing that knowing it from the description.  

5. Fig 1 caption: "the model includes ... four protein complexes (PSII, PSI, cb6f, ATPase)". But the Table includes more (e.g. terminal oxidase, RuBisCO, are these not protein complexes?)

6. Line 89: "Readouts include all intermediate metabolites". Better add "shown in Fig 1", because the model does certainly not include "all intermediate metabolites" (e.g. CBB is lumped into a single reaction)

7. Line 404: "On the other hand, our simulated rates lay a factor two below the O2 production measurements which were used to fit the dynamics of carbon uptake (Fig 2H)". The sentence is confusing. How were O2 production measurements used to fit the dynamics of carbon uptake? (also weren't O2 production measurements predicted, and not used to fit?)

8. From the previous response: "I assume equation S49 is incorrect, right?" I meant equation S50 (or the numbering shifted). It is an ODE for CBBa, the term on the right side is positive (so CBBa would go to infinity). I assume it is not an ODE. 

Overall: The first submission was clearly insufficient. The manuscript has improved. In particular, the (incorrect) claims about predictive power have been toned down. I still think there are a number of shortcomings (e.g. no error bars). In my opinion, the model also failed at almost all encounters with data, and had to be rescued by additional modifications (a light conversion factor, a changed description of NDH-1, etc ...) and all that while still staying within the single dataset used for the original parameterization. Also, the lack of acclimation (changes in expression of pigments, RuBisCO, etc ... which is largely unknown) will hinder actual predictions from the model. Therefore, the authors should remain careful in the interpretation.

**Have the authors made all data and (if applicable) computational code underlying the findings in their manuscript fully available?**

Reviewer #2: Yes

PLOS authors have the option to publish the peer review history of their article (what does this mean?). If published, this will include your full peer review and any attached files.

Reviewer #2: No

Figure Files:

Data Requirements:

Reproducibility:

References:

---

## [Editor Report · Decision Letter 3]

29 Aug 2024

Dear Mr. Pfennig,

We are pleased to inform you that your manuscript 'Shedding light on blue-green photosynthesis: A wavelength-dependent mathematical model of photosynthesis in *Synechocystis* sp. PCC 6803' has been provisionally accepted for publication in PLOS Computational Biology.

Best regards,

David Lea-Smith

Guest Editor

PLOS Computational Biology

Jason Haugh

Section Editor

PLOS Computational Biology

---

## [Editor Report · Acceptance letter]

6 Sep 2024

PCOMPBIOL-D-23-01193R3 

Shedding light on blue-green photosynthesis: A wavelength-dependent mathematical model of photosynthesis in *Synechocystis* sp. PCC 6803

Dear Dr Pfennig,

I am pleased to inform you that your manuscript has been formally accepted for publication in PLOS Computational Biology. Your manuscript is now with our production department and you will be notified of the publication date in due course.

With kind regards,

Anita Estes
